# Quantitative evaluation of large corporate climate action initiatives shows mixed progress in their first half-decade

**Ivan Ruiz Manuel** [1] ✉ **& Kornelis Blok** [1] ✉

Corporate climate initiatives such as the Science-Based Targets initiative and RE100 have gained significant prominence in recent years, with considerable increases in membership and several ex-ante studies stating how they could bring substantive emissions reductions beyond national goals. However, studies evaluating their progress are scarce, raising questions on how members achieve targets and whether their contributions are genuinely additional. Here we assess these initiatives by disaggregating membership by sector and geographic region and then thoroughly evaluating their progress between 2015–2019 using public environmental data disclosed by 102 of their largest members by revenue. Our results show that the collective Scope 1 and 2 emissions of these companies have fallen by 35.6%, with companies generally on track or exceeding scenarios keeping global warming below 2 °C. However, most of these reductions are concentrated in a small number of intensive companies. Most members show little evidence of emission reductions within their operations, only achieving progress via renewable electricity purchases. We highlight how intermediate steps regarding data robustness and implementation of sustainability measures are lacking: 75% of public company data is independently verified at low levels of assurance, and 71% of renewable electricity is obtained through low-impact or undisclosed sourcing models.

Non-state actors, which can be businesses, sub-national entities (cities and regions), and non-governmental organisations, have been posed as having the capacity to deliver emission reductions beyond the goals established during international climate negotiations[1–4]. Research into the environmental benefits generated by these actors has become a topic of increasing interest in climate change governance due to their involvement in global climate conferences[5,6]. Several studies have estimated their ex-ante potential for mitigation, with commitments ranging between 1.0–2.0 $GtCO_{2e}$ of mitigation beyond Nationally Determined Contributions (NDCs) by 2030[7,8], and best-case predictions of future growth in the transnational climate initiatives formed by these actors reducing up to 18-21 $GtCO_{2e}$ by 2030[9].

However, there is a lack of clarity on how actors in these initiatives implement their targets and whether substantive progress is made towards meeting them. Ex-post evaluations of transnational climate initiatives remain scarce due to complexities in gathering and assessing data[4,10]. In particular, most datasets accounting emissions usually stop at the national level, and voluntary disclosure platforms tend to have incomplete information or are difficult to assess[11]. Qualitative evaluations of transnational climate initiatives have highlighted how other issues exacerbate data problems, such as lack of institutional capacity due to inadequate staffing or funding[3], or lack of quantifiable targets[12]. The capacity of transnational initiatives to induce additional GHG emission reductions or renewable energy installations remains a contentious topic due to the previously mentioned problems, raising

[1]Faculty of Technology, Policy and Management, Delft University of Technology, Delft, The Netherlands. ✉e-mail: ivanruizmanuel@gmail.com; K.Blok@tudelft.nl

concerns of double-counting, effort fragmentation and emission leakage effects[4,6].

In the case of companies, evaluation is complicated further by a lack of convergence in how data is presented, driven by a plurality of reporting standards, disclosure platforms, legislative differences and conflicting stakeholder priorities[13–15]. Widely used reporting standards such as the Global Reporting Initiative (GRI) leave leeway on how environmental information is disclosed, and centralised sustainability reporting platforms such as CDP (previously the Carbon Disclosure Project) have incomplete submissions or are affected by year-by-year inconsistencies[10,16]. This lack of convergence means that, for the same company and year, the information in a CDP response and a public sustainability report might not be comparable due to data omissions, differing calculation boundaries, or incompatible presentation (e.g., disclosing emissions in absolute numbers or as mere percentage decreases).

The few available ex-post analyses give mixed results, with a recent report stating that 80% of the targets set by 119 companies are on track to be overachieved[17], and Giesekam et al.[18] showing that 74% of primary targets in 81 companies are either on-track or surpassed. In contrast, a recent analysis of 25 companies with net-zero targets showed low transparency and target integrity levels, omissions in value chain emissions, use of low-quality carbon offsets, and poor renewable energy purchasing practices[19]. Similarly, another study estimated that nearly half of the energy purchasing commitments of 115 companies would not lead to additional renewable generation capacity[20]. However, most studies still quantify progress as a percentage, making overall mitigation uncertain due to the varying size and characteristics of the companies included. Disclosing by the initiatives themselves also has drawbacks as individual emissions or energy consumptions are undisclosed, leaving the contributions indistinguishable without consulting secondary sources[21–23].

This study assesses GHG emission reductions and other relevant metrics in two major corporate climate initiatives: the Science-Based Targets initiative (SBTi) and RE100. The SBTi has the goal of evaluating and approving GHG emission reduction targets set by companies. This is done by comparing absolute emission reduction trends against Integrated Assessment Model scenarios that keep global emissions at least below 2 °C or 1.5 °C (Absolute Contraction Approach[24,25]), or by comparing sector-specific emission intensities against scenarios that keep global warming below 2 °C (Sectoral Decarbonisation Approach[26]). RE100 aims to promote renewable electricity use in companies, requiring its members to reach 100% renewable electricity by 2050 at the latest[27]. Both initiatives have shown high mitigation potential in ex-ante studies[28,29], with Lui et al.[9] estimating that, if each initiative grew to 2000 members by 2030 and kept a similar company composition, the SBTi and RE100 could mitigate 2.7 and 1.9–4.0 GtCO$_{2e}$, respectively. However, these amounts may not manifest due

to overlaps in membership, which already occur[21], or if newer members are less emission-intensive.

We evaluate these initiatives in three steps. First, we use Fortune's Global 500 list (G500)[30] to identify their largest members by revenue. We then group participating and nonparticipating companies by region using the location of their headquarters, and by energy sector using industry classification standards. Second, we collect relevant environmental data of participating companies using CDP responses and public reports, which is then validated to reduce disclosure discrepancies and to ensure it complies with requirements set by the GHG Protocol standard[31] and CDP guidelines[32]. Third and last, we use the logical framework developed by ref. 11 to descriptively assess participants using four indicators: the ambition of their targets, the robustness of their disclosure practices, the changes implemented to achieve targets, and the substantive impact that this had on their emissions and share of renewable energy.

The study only covers direct emissions produced within company boundaries (Scope 1)[33], and indirect emissions from energy purchases (Scope 2)[34]. We do not evaluate targets related to emissions from upstream and downstream activities (Scope 3)[31]. Although Scope 3 emissions are often larger than the other two, the obligatory primary targets of both the SBTi and RE100 solely cover Scope 1 + 2. Only SBTi members whose Scope 3 emissions account for more than 40% of their total emissions are required to set secondary targets for this scope[35], and in some cases, they only require enrolling suppliers in the initiative[36]. Generally, we consider that changes within the operational or energy purchasing behaviour of these companies gives information on the nature of their commitment.

In this work, we show that 102 members of these two initiatives reduced their collective emissions by 35.6% between 2015–2019. SBTi members exceeded their aggregated GHG emission reduction targets, and RE100 companies showcased an average annual growth of consumed renewable electricity of 31.2%. However, our intermediate indicators revealed a general lack of high-quality verification of the environmental data, and a decreasing level of transparency in how renewable energy is sourced. Most of the mitigation achieved within company boundaries is concentrated in eight emission-intensive companies exclusively participating in the SBTi. All other companies, including the entire RE100 sample, only achieved indirect emission reductions by purchasing renewable energy, 71% of which is sourced through unknown or low-impact instruments. We conclude this paper by calling for consistent and transparent disclosure processes[36], and increased attention into how renewable energy is sourced by members.

## Results
### Characterising participation

A total of 137 companies in the G500 have enrolled in the SBTi and RE100 initiatives, representing a combined 9.22 US$Trillion in revenue (28% of the total revenue in the G500). Of these, 116 participate in the SBTi, comprising 9.6% of the initiative's total members (1205 as of February 2021). This subset includes companies with approved targets and those committed to setting them (Table 1), with a higher ratio of target approval (62%) than the initiative as a whole (49%). However, approved targets have a similar distribution of qualifications: 53% in the 1.5 °C category and 25% in well-below 2 °C (compared to 51% and 26%, respectively, in the whole initiative), with the remaining targets in the obsolete 2 °C qualification. For committed members, SBTi specifies a 2-year limit to get targets approved and published before a company is removed from their listings[35]. At the time of data collection, 24 out of the 44 committed members had already exceeded the time limit, 14 of which were companies in the financial sector. RE100 membership was lower at 68, all with targets set, representing 24% of the initiative's membership (289 as of February 2021). An overlap of 48 companies

**Table 1 | Summary of G500 companies in the SBTi**

| Qualification | n | Absolute target | | | Intensity target | | |
|---|---|---|---|---|---|---|---|
| | | Scope 1 | Scope 2 | Scope 3 | Scope 1 | Scope 2 | Scope 3 |
| 1.5 °C | 38 | 35 | 35 | 28 | 2 | 1 | 9 |
| Well-below 2 °C | 18 | 15 | 15 | 9 | 3 | 3 | 5 |
| 2 °C | 16 | 15 | 15 | 10 | 3 | 2 | 5 |
| Committed | 44 | – | – | – | – | – | – |
| Total | 116 | 65 | 65 | 47 | 8 | 6 | 19 |

Grouped by target qualification and GHG Protocol scope covered, and subdivided into absolute and intensity targets, if applicable. Scope 1 and Scope 2 may be covered by a single target (e.g., a combined reduction of 40%); Scope 3 targets may cover all or only some of the 15 emission categories in this scope[31].

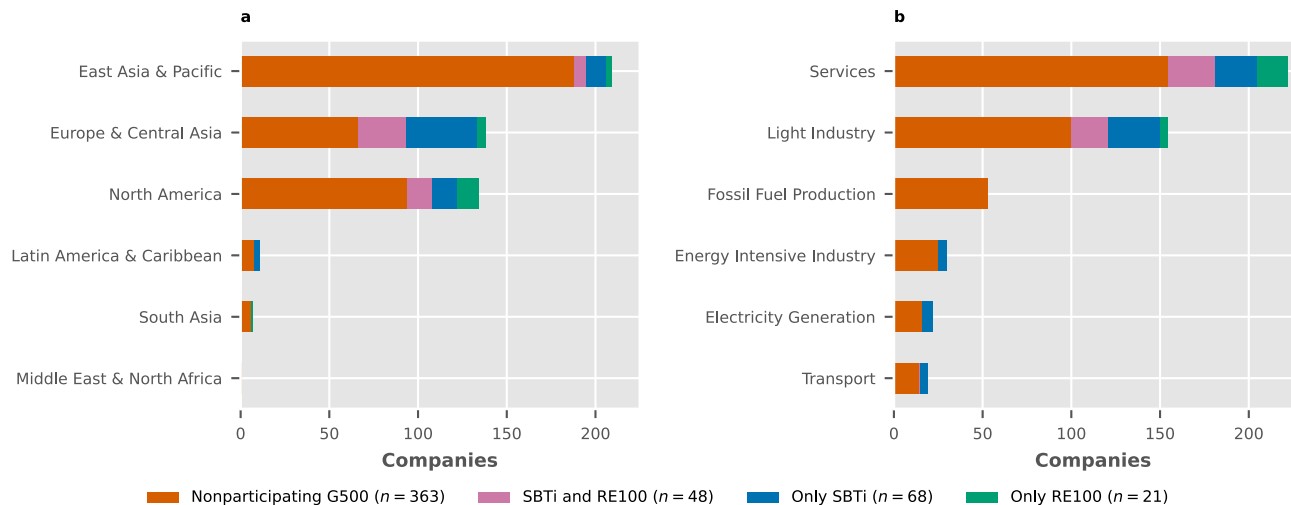

**Fig. 1 | Geographical and sectoral distribution of the initial sample of companies.** Subdivided into nonparticipating Fortune Global 500 (G500) companies and those enrolled in the Science-Based Targets initiative (SBTi), RE100, or both.

**a** Distribution by World Bank regions (missing regions have no companies in the ranking). **b** Distribution of participants per energy sector (see Supplementary Table 2 for definitions).

participated in both initiatives, with 33 of them having approved targets in the SBTi.

Regional engagement with the initiatives was identified through geographical analysis (see Fig. 1a). The results showed that high-income European economies had the highest levels of engagement, with 47 out of 95 in the EU27, 15 out of 22 in the UK and 9 out of 14 in Switzerland. Australia was the only non-European country with high engagement (3 out of 5). Medium engagement (between 33–25%) corresponded to the Americas: the U.S. was the highest at 40 out of 121, followed by Brazil (2 out of 7) and Mexico (1 out of 4). Japan was the only nation outside the Americas with similar percentages (16 out of 53). Low engagement (≤20%) was primarily seen in Asia, with Chinese companies mostly absent (1 out of 124) despite previous studies on transnational climate initiatives finding some degree of engagement within the country[1]. Other Asian countries with low participation include South Korea (0 out of 14), Taiwan (1 out of 9) and India (1 out of 7). However, some high-income nations with low participation stood out: Canada (0 out of 13), the Netherlands (2 out of 11) and Italy (1 out of 6). Africa is absent from the G500, so our analysis could not identify members in this continent.

We also evaluated the distribution of members in relation to key energy sectors (Fig. 1b). The results showed that Light Industry and Services had the highest number of participants in the initiatives (54 out of 154 and 67 out of 222, respectively). However, only the SBTi had members in emission-intensive sectors such as Electricity Generation (6 out of 22) and Energy Intensive Industry (5 out of 30). RE100 had 17 exclusive members in the Services sector, 15 of which were financial businesses with comparatively low energy use. Exclusivity in this initiative was rare in other sectors. None of the companies in the Fossil Fuel Production sector participated in either initiative. It is important to note that at the time of this study, SBTi only allowed these companies to commit to the initiative, but did not formally approve any of their targets. Additionally, RE100 explicitly disallowed energy producers from becoming members[27].

## Boundaries of the logical framework analysis

Our evaluation only covers emissions and energy generated within company boundaries (Scope 1)[31] and emissions generated through energy purchases (Scope 2)[34]. We excluded targets covering the supply chain and downstream activities of members (Scope 3) from our study. We analysed data from 2015 to 2019, the last year for which complete CDP and sustainability report data was available. We included RE100

participants who joined at or before 2019 (n = 58) and SBTi members whose target baseline was set at or before 2019 (n = 70), while excluding members without approved targets and a few companies that did not disclose performance data with sufficient transparency (see Supplementary Information). We did not project target trends for seven SBTi companies with intensity targets, as these targets require extra metrics of company activity (e.g., tons of cement or kWh generated per year) and include additional disclosure complexities[18]. Overall, our logical framework evaluation includes 102 businesses, 26 of which participated in both initiatives.

## Ambition indicators

For SBTi, our analysis only covers 63 members with absolute targets (i.e., aiming to reduce GHG emissions directly) and excludes seven members with intensity targets. We projected annual Scope 1 + 2 emissions assuming they stay constant before the target's baseline year or beyond the target's final year. These trends were compared against normalised scenarios for OECD + EU nations featured in IPCC reports[37] (see Fig. 2). Results showed that collective efforts in our sample are consistent with shared socioeconomic pathways keeping mean temperature increase below 1.5 °C[38,39], with an overall reduction of 43% by 2030 (from a 298 $MtCO_{2e}$ 2015 baseline). The Services and Light Industry sectors (n = 56), which represented 46% of target baseline emissions in 2015, remained within 1.5 °C compatible trends by 2030[40], with a combined reduction of Scope 1 + 2 emissions of 49%. However, the few Energy Intensive Industry and Electricity Generation companies (n = 5), which encompassed 50% of baseline emissions in 2015, only aimed for a combined reduction of 37%, which marginally falls below the projected threshold to be in line with 1.5 °C.

For RE100 (n = 58, Fig. 3), target trends were also projected and compared against 1.5 °C scenarios for OECD+EU nations. This group of companies could reach a share of 94% renewable electricity by 2030 (148 TWh total), exceeding the global share projected by scenarios keeping global warming below 1.5 °C[40], and also exceeding the required ratio for OECD + EU nations[37], where most of these companies have their headquarters. However, there are noticeable differences between the two participating energy sectors: Services (n = 38) aim for a collective 99% by 2030, while companies in the Light Industry sector only aim to exceed 85% by the same year.

Although most of these companies are headquartered in OECD countries, the distribution of their operations might differ. To account

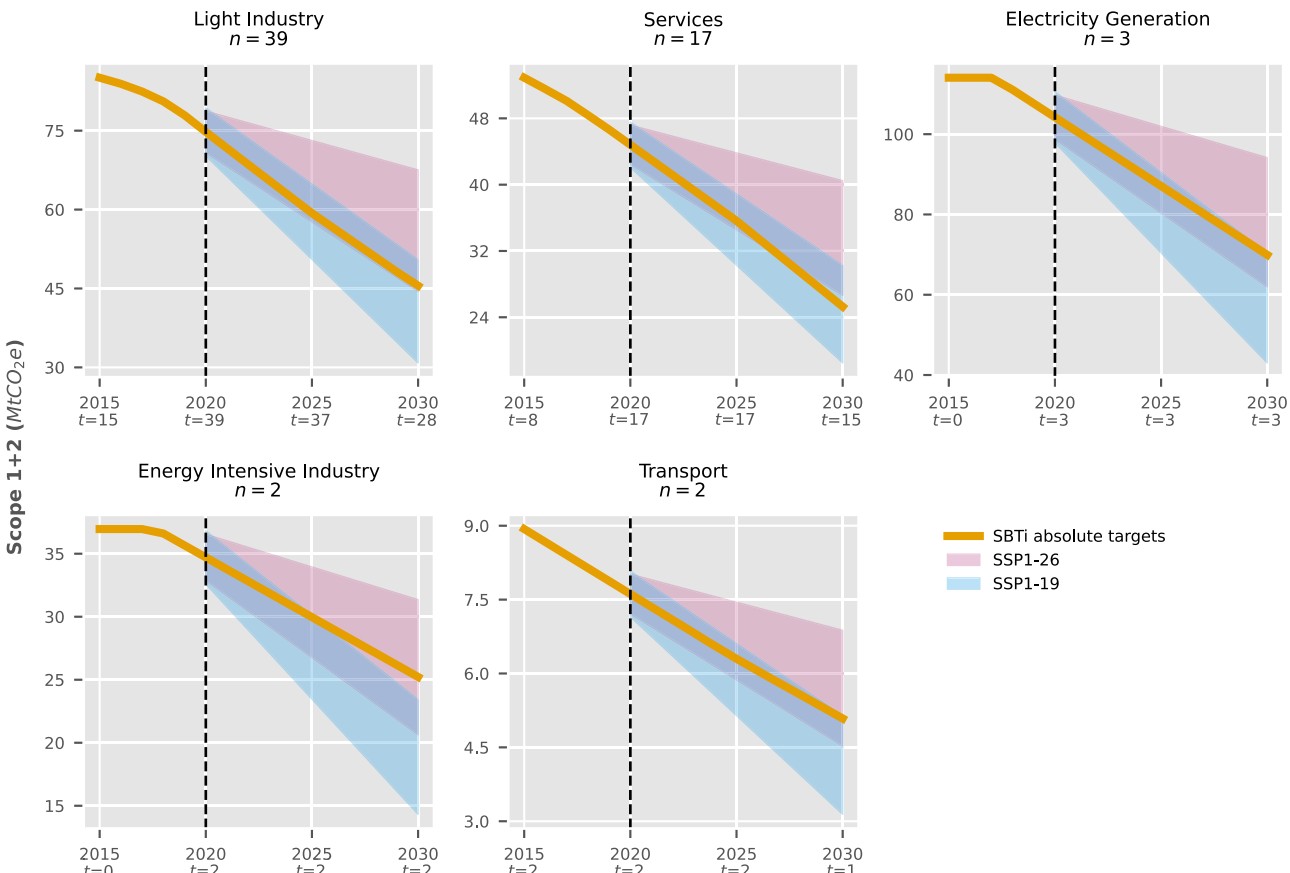

**Fig. 2 | Ambition of companies with absolute targets in the Science-Based Targets initiative (SBTi) compared to regional scenarios.** Grouped by sector. Targets were compared against regional scenarios under Shared Socioeconomic Pathways with low challenges to mitigation and adaptation, keeping global warming below 1.5 °C (SSP1-19) or 2 °C (SSP1-26). All scenarios aggregate nations in the Organisation for Economic Co-operation and Development plus European Union members and candidates (R5OECD90 + EU). $t$ represents the number of companies with active targets in a year.

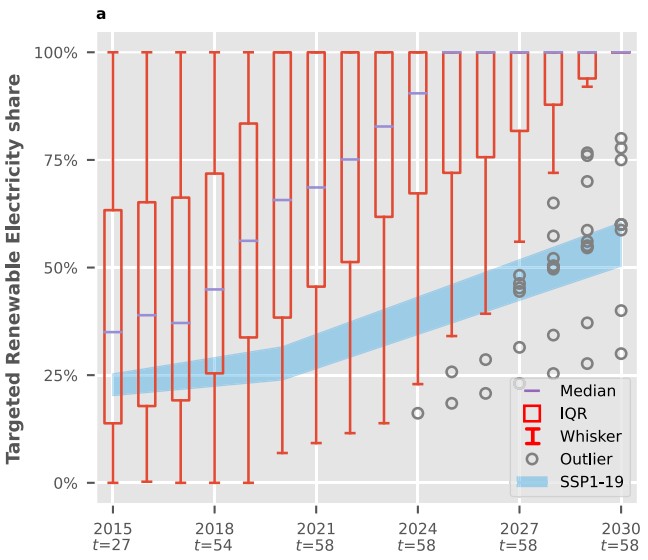

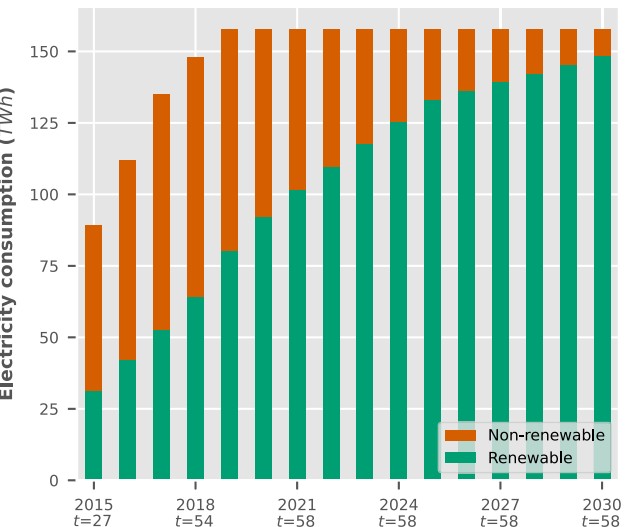

**Fig. 3 | Ambition of companies in RE100 ($n = 58$) compared to OECD scenarios.** Members without approved targets were excluded in the initial years. $t$ represents the number of companies with active targets each year. **a** Renewable electricity ratios targeted by each company compared against scenarios under Shared Socioeconomic Pathways with low challenges to mitigation and adaptation keeping global warming below 1.5 °C (SSP1-19). The scenarios only include nations in the OECD and EU members and candidates. Boxes envelop the interquartile range (IQR) of the data with the median as a line (25th, 50th and 75th percentiles). Whiskers stretch from the box by 1.5x of the IQR. Values outside this range are shown as outliers. **b** Growth of renewable electricity consumption in the initiative if targets are met, assuming total consumption remains constant after 2019.

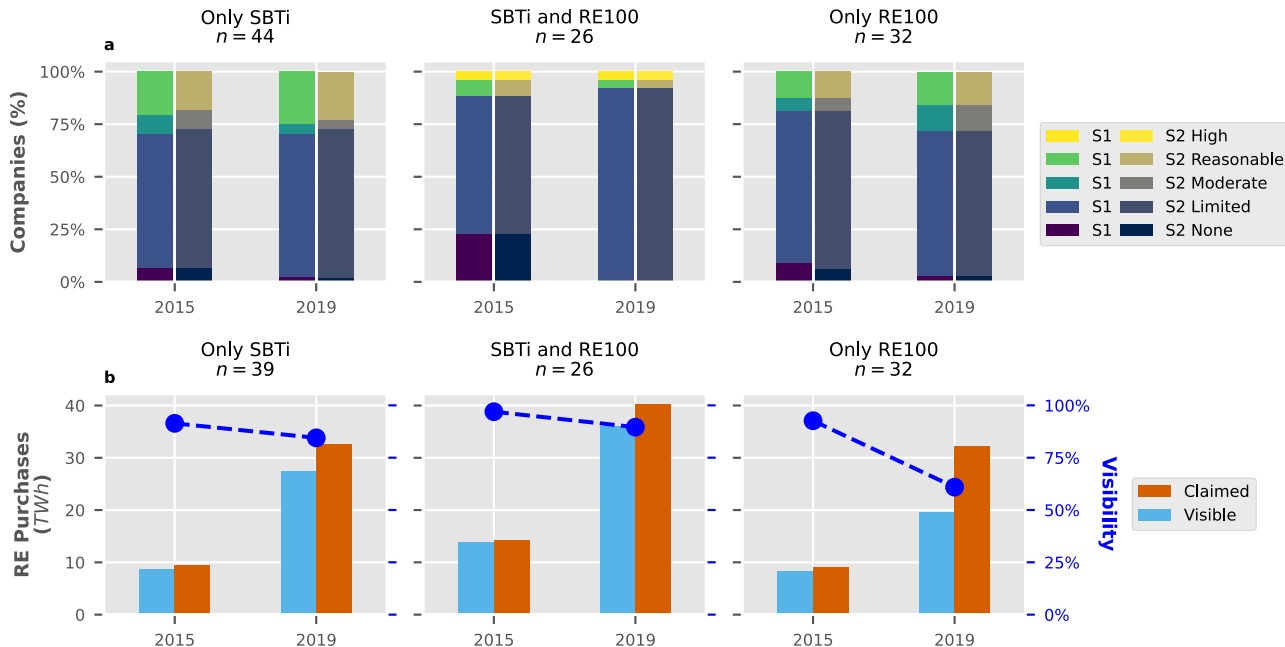

**Fig. 4 | Robustness indicators for companies with targets set.** Separated into companies exclusively in the Science-Based Targets initiative (SBTi), those only in RE100 and companies enrolled in both. **a** Ratio of third-party verification of Scope 1 (S1) and Scope 2 (S2) emissions as classified by CDP[41]. **b** Comparison of claimed purchased renewable energy (RE) against purchases with a publicly visible sourcing model. Companies in the Electricity Generation sector were omitted due to issues in their disclosed data (see Methods).

for this, we compared targets against global scenario trends (see Supplementary Figs. 3 and 4). Doing so extends the allowable range for 2 °C compatibility in the models selected for this study (see Methods) and reduces the acceptable upper range for 1.5 °C. Most RE100 members still outperform 1.5 °C conditions due to the initiative's requirement of 100% renewable electricity by 2050. However, SBTi is no longer collectively within 1.5 °C due to an excess of 18.2 MtCO$_2$e. Services was the only sector that maintained 1.5 °C trends, with all others only reaching below 2 °C ambition.

**Robustness indicators**

In this study, we define robustness as the degree of oversight and transparency of environmental data released by corporations. More specifically, we examine the type of third-party verification used by companies and whether the model used to purchase renewable energy is visible to the public.

CDP accepts various standards and levels of third-party verification, which are grouped into five categories: None, Limited, Moderate, Reasonable, and High[41]. Limited and Moderate assurance can generally be considered equivalent and involve a narrower scope of evaluation, such as inquiries or document analysis[42–44]. In contrast, Reasonable and High verification, which are also generally equivalent, are more costly[13], involve more scrutiny, and result in a positive statement of opinion by the auditor. Limited and Moderate verification do not typically result in such statements (see the Supplementary Material for more information on assurance equivalency).

Companies showed similar verification trends regardless of whether they joined one or both initiatives (Fig. 4a), with the majority choosing to do so only at a Limited level. By 2019, 75% of the participants still verified their data at a lower level of scrutiny. This behaviour persisted even in emission-intensive sectors: no Energy Intensive Industry was verified at Reasonable or High level ($n = 3$), and the only Electricity Generation companies that did so ($n = 3$, 5 in total) are headquartered in countries with legal requirements to do so[45]. We also identified other trends: first, the same level of assurance is provided for Scope 1 and 2 emissions, apart from some exceptions ($n = 2$). Second,

the share of companies who did not seek external verification decreased from 12% to 2% in our sample of 102.

Visible renewable energy purchases refer to those where companies publicly disclose the sourcing model used to obtain renewable energy. As shown in Fig. 4b, visibility has decreased as the quantity of claimed renewable energy has increased, but at varying levels between both initiatives. From 2015 to 2019, the total claimed renewable purchases increased from 33 to 105 TWh, and while visibility increased in absolute terms, it decreased in relative terms from 94 to 79% ($n = 97$). This decline is mainly due to recent changes in CDP questionnaires which barred Financial companies from disclosing their purchasing models[46]. This explains why the decrease is primarily seen in RE100, since it has a high amount of members in the Financial sector ($n = 22$) which reached only 21% visibility in 2019 out of the 8.5 TWh they claimed. Although RE100 requests its members to report through different methods in such cases[47], this information may not be publicly available.

**Implementation indicators**

The progress made in implementing the targets is measured by analysing changes in the energy profile of SBTi members, which are categorised as either energy producers or end-users, as well as by examining the preferred model for sourcing renewable energy among RE100 members.

In the case of SBTi members in the Electricity Generation sector ($n = 5$), limitations in early CDP questionnaires and missing data in public reports meant that net energy generation data could only be completed for 2017–2019. These companies already had a high percentage of low carbon energy generation (i.e., renewable and nuclear), which increased from an average of 49 ± 31% in 2017 (1255 TWh total) to 59 ± 35% in 2019 (1156 TWh total), with a decrease in fossil fuel generation and an increase in renewable generation as the main drivers.

For end-use SBTi members ($n = 65$), we distinguished between internal energy use related to Scope 1 (fuel use and non-fuel-based renewable self-generation) and external energy use attributed to

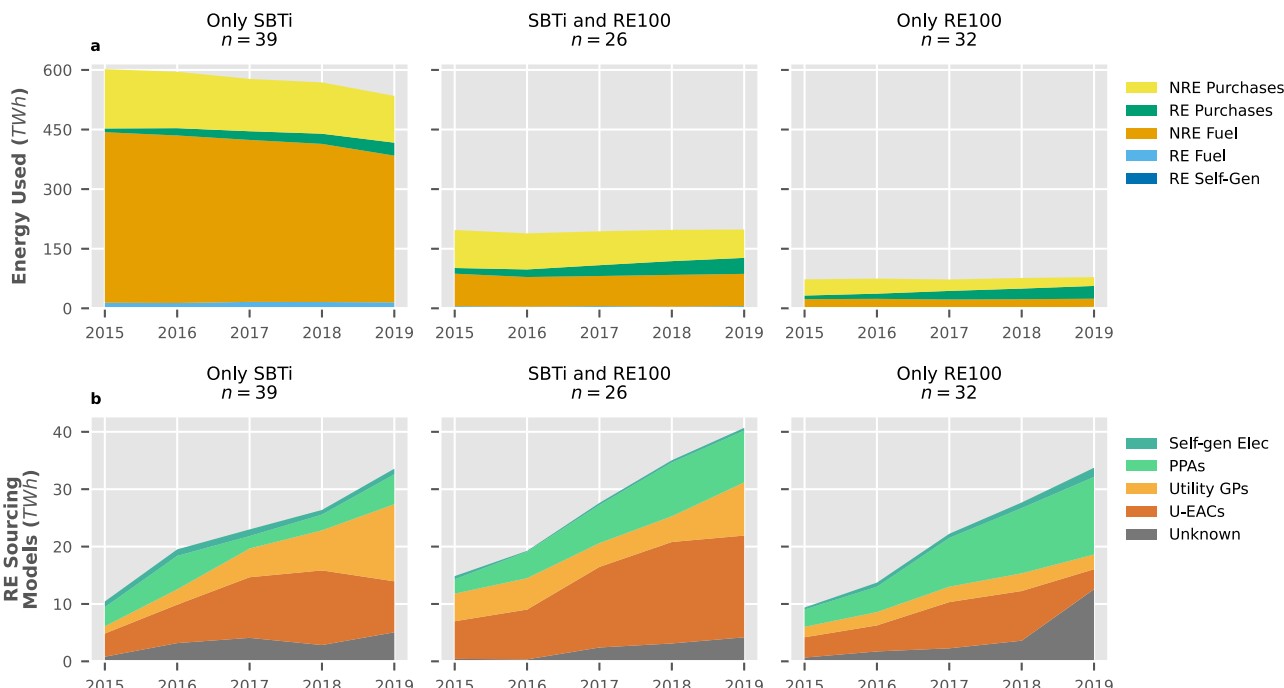

**Fig. 5 | Implementation indicators for energy end-use companies. a** Trends in the use of renewable (RE) and non-Renewable (NRE) fuels used within company boundaries and purchased electricity, heat, steam and cooling. **b** Preferences in the model used to source non-fuel-based renewable energy, ordered from low additionality, such as Unbundled Energy Attribute Certificates (U-EACs) and Utility Green Premiums (Utility GPs), to high additionality, such as Power Purchase Agreements (PPAs) and renewable self-generation (Self-Gen); Unknown represents the difference between claimed and visible purchases (see Fig. 4b).

Scope 2 (purchased energy)[31]. Internal energy use decreased from 530 to 471 TWh between 2015–2019, with 97% of this change occurring in the Energy Intensive Industry sector alone (*n* = 3). This small group of companies was responsible for 60% of all internal energy end-use in 2015. Consumption of renewable fuel and non-fuel-based renewable self-generated energy did not display a substantial trend of increment, accounting only for a combined 21 TWh in 2019. Externally sourced energy had two trends: a slight decrease in overall consumption from 269 to 262 TWh between 2015–2019, and an increase in purchased renewable energy from 9% to 28%, with all sectors substantially increasing their use. Heavy polluters reduced fossil fuel use in their internal energy processes, possibly through efficiency measures. Less intensive sectors primarily focused on increasing their renewable electricity purchases, an expected external change. Detailed statistics on the data can be found in Supplementary Table 4.

We evaluated energy purchases by comparing the sourcing models employed to obtain renewable electricity: Unbundled Energy Attribute Certificates (U-EACs), Utility Green Premiums (Utility GPs), Power Purchase Agreements (PPAs) and self-generated renewable electricity. High-quality models such as PPAs, which are long-term contracts between companies and independent producers, and self-generation[19] are becoming more prevalent in the initiative, growing from 26% to 33% between 2015–2019, with PPAs being the most popular of the two accounting for 92% of all high-quality sourcing in 2019. The literature considers U-EACs and Utility GPs to have lower quality[19,48,49] meaning they may not translate to actual displacements of fossil fuel energy sources and GHG emission reductions. U-EACs can be purchased separately from a company's energy consumption, and are considered poor alternatives due to their low price[50], weak relation to additional renewable energy installations[51] and because they do not reflect the physical flow of energy at the point of consumption[20]. Utility GPs are contractual instruments between utilities and companies which offer lower additionality due to the plethora of government support schemes

offered to utilities to increase their renewable generation and, in some cases, because they may be based on repackaged U-EACs purchased by the utility[48].

In the case of RE100 (*n* = 58), trends indicate that U-EACs and Utility GPs are becoming less popular, decreasing from a combined 69% to 44% of all sourced TWh. However, the low transparency of Financials (*n* = 22) in 2019 might be obscuring higher low-quality sourcing model usage. Assuming these companies kept their 2018 preferences (84% low-quality usage), the low-quality ratio in the initiative ends up at 52% for 2019.

A clear relationship between implementation metrics for both initiatives can be drawn, as most end-users in the SBTi opted to purchase renewable energy instead of reducing their internal fossil fuel use (Fig. 5a). Excluding Electricity Generation companies, SBTi-only members (*n* = 39) consumed 535 TWh of energy in 2019 and were the only group were total energy consumption decreased. In the same year, overlapping members (*n* = 26) consumed 198 TWh, and RE100 exclusive members (*n* = 32) only consumed 79 TWh, primarily due to the high concentration of Service companies in this group (*n* = 27). However, the total sourced renewable energy is similar in the three groups (Fig. 5b), with RE100 being a clear determinant of better practices: the RE100 exclusive group had the highest share of high-additionality sourcing models at 46%, while its SBTi counterpart had the lowest at 18% and overlapping members had a slightly better 23%. However, collective sourcing model indicators remained poor in 2019: 71% of all the renewable energy purchased had either low additionality or remained undisclosed.

**Substantive progress indicators**
Overall, companies in both initiatives reduced their collective GHG emissions between 2015–2019. Assuming no overlaps between utility companies and other members, SBTi participants with absolute targets (*n* = 63, see Fig. 6a) reduced their GHG emissions at an average of −7.8% per year and surpassed their targeted reductions by 34 MtCO$_{2e}$ in 2019. All sectors except Transport (*n* = 2) show this trend, with Electricity

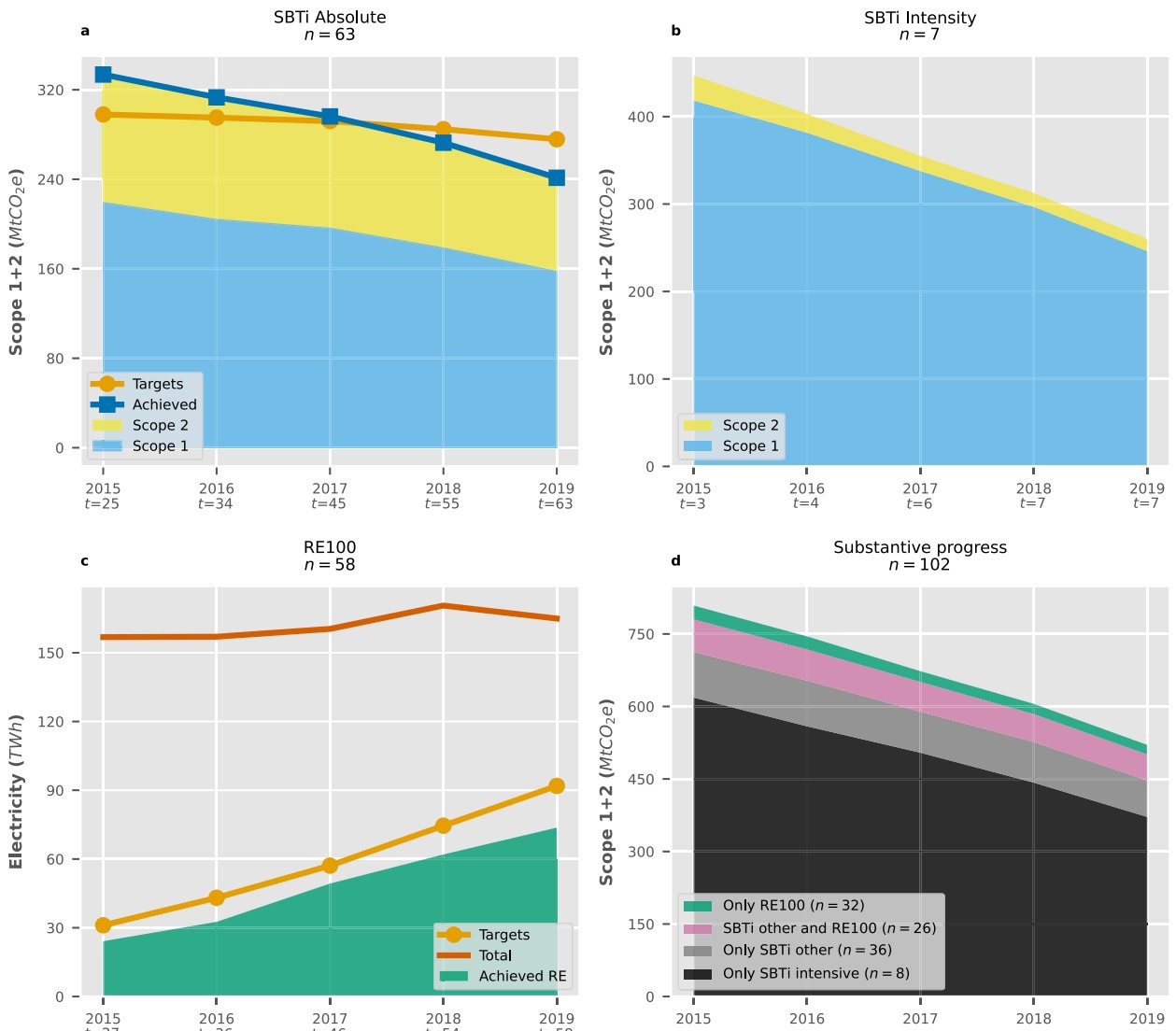

**Fig. 6 | Substantive progress indicators for different target types, where *t* represents the number of companies with active targets in a year. a** Trends absolute targets in the Science-Based Targets initiative (SBTi). Target trends assumed constant company emissions before the baseline year to ensure comparability against historic emissions. **b** Trends in SBTi members with intensity targets. **c** Renewable electricity (RE) versus estimated trends in RE100 members (target trend assumes no renewable electricity use before the baseline year to ensure comparability against historic trends). **d** Combined Scope 1 + 2 emissions of all companies subdivided by membership and keeping SBTi members in emission intensive sectors separate (i.e., Electricity Generation, Energy Intensive Industry). Scope 2 Market Based data was preferred if available. See Supplementary Tables 6 and 7 for more information on performance per GHG Protocol Scope.

Generation companies ($n = 3$) being responsible for 78% of the over-achievement. However, collective 2015 emissions exceeded the sum of all absolute target baselines by 36 $MtCO_{2e}$. This suggests that overall emission reductions achieved in this group may be a continuation of previous trends rather than an acceleration of mitigation action. A similar trend can be seen in SBTi members with intensity targets ($n = 7$, see Fig. 6b), which showcased steady emission decreases at an annual average of −12.6%. As a whole, Energy-intensive members appear to play a crucial role in the initiative's achievements: grouping companies in the Electricity Generation ($n = 5$) and Energy Intensive Industry ($n = 3$) sectors gives a collective pace of −11.9% per year, while the rest ($n = 62$) only reach −5.4%.

In the case of RE100 ($n = 58$, see Fig. 6c), several companies over-reported their renewable electricity usage to RE100 in the early years of the initiative, affecting target baselines in 2015 and producing an offset between targeted renewable consumption and actual renewable sourcing of 19 TWh by 2019. Despite this, progress remains adequate: renewable electricity use has grown at an average of 31.2% per year,

which is close to the targeted 32.7%. The initiative reached 74 TWh of renewable electricity consumption in 2019, meaning 45% of total electricity comes from renewable sources.

The combined action of both initiatives ($n = 102$) decreased their Scope 1 + 2 emissions by 288.2 $MtCO_{2e}$ between 2015–2019, repre-senting a 35.6% reduction from their baseline of 808.7 $MtCO_{2e}$ (see Fig. 6d). However, sectoral disaggregation veers progress heavily in SBTi's favour: 86% of total reductions were due to exclusive members in the Electricity Generation and Energy Intensive Industry sectors ($n = 8$), accounting for approximately 98.6% of all internal Scope 1 emission reductions (see Supplementary Table 5). Other sectors, including all RE100 participants, only show evidence of emission mitigation occurring outside their operational control (i.e., Scope 2). While this focus aligns with the goals of RE100 and can be expected as a first step of mitigation, it is critical to note that the effectiveness of this approach depends on the quality of the sourcing model used to acquire renewable electricity, as low-quality models are unlikely to result in additional renewable installations that displace fossil-based

generation. Unfortunately, our implementation indicator suggests that there is still room for improvement in this regard.

## Discussion

The number of corporations participating in transnational climate initiatives is increasing rapidly, with many publicly disclosing climate targets or expressing their intent to do so soon. To add to literature on non-state action, we conducted a validated quantitative analysis of environmental data publicly disclosed by the largest members of the SBTi and RE100, two corporate climate initiatives that have high potential for mitigating climate change[9]. Our analysis covers four areas: the ambition of company targets (compared to scenarios that keep average temperature increase at least below 2 °C), data robustness (use of third-party verification and transparency in renewable energy sourcing), the implementation of targets (changes in energy use and renewable electricity purchasing preferences), and substantive progress towards meeting targets (reductions in Scopes 1 + 2 GHG emissions and increments in the share of renewable electricity consumed). We also contextualise these initiatives by grouping members according to their geographical distribution and energy sector. While it was not possible to evaluate all the companies participating in these initiatives, our sample of 102 companies likely accounts for 59% of the 1.2 GtCO$_{2e}$ coverage mentioned in SBTi reports for 2019[22] and 59% of the 289 TWh of electricity consumed by RE100 members in the same year[21]. Companies not included were either too small to be ranked among G500 companies or were excluded due to limited data disclosure.

Results indicate that company targets are aligned with shared socioeconomic pathways for below 2 °C[52]. The ambition of SBTi's participants corresponds to emission pathways keeping global warming below 2 °C, while RE100 members' targets surpass the 1.5 °C requirements for the share of renewable electricity. These trends show that members adhere to the target-setting approaches required by each initiative, even if they vary in complexity. This translated into a 35.6% decrease in combined Scope 1 + 2 emissions between 2015–2019 (288.2 MtCO$_{2e}$ abated). This was driven by reduced fossil fuel use in electricity producers and an increase in renewable energy purchases by energy end-users, with renewable electricity accounting for 95% of the increment. This covers 44% of the GHG reductions by company initiatives estimated by an early ex-ante study for 2015–2020[28], which omitted the RE100, had a much lower membership for the SBTi and included several other initiatives.

However, most emission reductions were achieved by a limited number of utilities and energy-intensive companies, with 86% of GHG reductions attributed to just eight SBTi members. Sectors with high levels of participation, such as Services and Light Industry, showed little evidence of emissions reductions within their operational boundaries, with most progress being in Scope 2 mitigation through contractual changes in their energy sourcing. These trends are unsurprising: electric utilities are expected to be the main contributors towards GHG mitigation by 2030[53], and changes in energy purchases are easy first-step gains for energy end-users.

By 2019, the RE100 initiative had only covered 14.2% of the combined total emissions, and a mere 3.8% if only exclusive members are accounted for. This limited progress can be attributed to the initiative's specific focus on 100% renewable electricity consumption and a lack of participation from energy-intensive industries. This suggests that the mitigation potentials of these initiatives are not similar contrary to what some ex-ante studies suggested[9,29]. However, we observed that RE100 members had better renewable energy sourcing practices, with companies exclusively participating in the SBTi sourcing considerably less renewable energy through high-quality models such as PPAs and renewable self-generation. Both initiatives should prioritise pushing their members towards internal change and better sourcing practices since U-EACs, the preferred method for claiming

the use of renewable electricity, do not promote new renewable investments due to their low prices and show limited evidence of achieving real-world emission reductions[20,51]. Here, collaboration and sharing of best practices, evaluation methods and performance data could be crucial.

Other aspects of disclosure also need improvements. Even if the use of third-party verification has increased, most companies only verify at a low level, and there is no evidence of a trend towards stricter verification regardless of sector. Although complexity and cost are often cited as barriers towards more thorough evaluation[11,13], third-party verification is often the only method that the public and these initiatives have to assess company claims. In addition, there is also an increasing lack of transparency regarding renewable energy purchases, as well as issues that obstruct the appropriate use of CDP data, such as recurrent guideline updates complicating longitudinal comparisons[10,16] and active omissions by the companies themselves[19,54]. These problems prevent higher levels of scrutiny and will reduce the accuracy of future evaluations. Improved, centralised and converging ways of disclosure should take priority, as transnational climate initiatives are not exempt from attempts at greenwashing[15,19,55,56], exemplified by how several members of the SBTi have been removed in the past due to lack of genuine commitment[57].

Given the weak alignment of national policies with 1.5 °C pathways[58], transnational climate initiatives such as the SBTi and RE100 will likely remain an important part of global efforts. Our study shows that companies are collectively successful in meeting the goals established within these two initiatives. However, achievements may not directly translate into additional GHG emission reductions or renewable energy capacity. Recent studies have shared concerns about the additionality of corporate climate action due to low transparency[19,36] and the lack of a robust mandatory reporting framework[18]. These issues were forewarned by studies expressing worries over the quality of initiatives outside the UNFCCC[5,6], so much so that earlier assessments of their potential for mitigation already assumed that about one-third of the emission reductions achieved would not be additional to NDCs[28]. Most companies in this study sourced their renewable electricity through models associated with low additional renewable capacity, corroborating other recent SBTi evaluations[20]. Similarly, 73% of the GHG emissions covered by RE100 overlapped with the SBTi. As a result, it can be expected that until 2019 the overlap with national pledges is higher than what was suggested by studies stating best-case predictions[9,29].

Quantifying the additionality of climate initiatives is not a straight forward process since, for example, lack of action by other actors can counteract their benefits at a national level[4]. To determine additionality more accurately, it is necessary to investigate in depth at the company level or compare participating and nonparticipating businesses across various sectors and regions, which are crucial steps to advance this field of research. Since our results showed that mitigation is heavily concentrated in a few members in high-emitting sectors, focusing on them might achieve the right balance between tractability and materiality. For initiatives like the SBTi and RE100, a crucial step is requiring high-quality renewable energy sourcing models, such as PPAs and self-generation. Similarly, more transparent disclosure and a better evaluation of accounting procedures are prerequisites for credible contributions by companies to global GHG emissions reduction[36]. Without these measures, evaluation of other metrics such as Scope 3 mitigation will remain difficult. Further research into Scope 3, including the fitness of target coverage and the robustness of methods used to calculate and verify emissions, could be crucial in understanding whether companies in initiatives positively influence the behaviour of other entities in their value chains.

Finally, it is noteworthy that membership is concentrated in OECD nations, which mirrors the results of previous studies looking into the distribution of transnational climate initiatives[1,5] and initiative

**Table 2 | Logical Framework used to evaluate the SBTi and RE100 in this study**

| Type of progress indicator | Benchmarks and baselines | Key metrics | Period |
|---|---|---|---|
| Ambition | Extended company targets (Scope 1 + 2) | Targets within 1.5 °C or 2 °C trends | 2015–2030 |
| | SSP1-19 scenarios (normalised) | | |
| | SSP1-26 scenarios (normalised) | | |
| Robustness | Third-party verification qualification | Share of Reasonable and High verification | 2015–2019 |
| | Claimed renewable energy purchases | High visibility into sourcing preferences | |
| | Transparent renewable energy purchases | | |
| Implementation | Non-renewable energy used or produced | Share of renewable energy | 2015–2019 |
| | Renewable energy used or produced | Share of high additionality sourcing models | |
| | Renewable energy sourcing model employed | | |
| Substantive Progress | Extended company targets (Scope 1 + 2) | Performance against targets | 2015–2019 |
| (direct impact) | Actual collective GHG emissions | Change in GHG emitted | |
| | Actual collective Renewable Electricity use | Change in renewable electricity use | |
| Evaluation of potential | Narrative description of company actions should coincide | All the indicators above | Ex-post review |
| | with effective and efficient mitigation approaches | | |

Adapted from[11], and subdivided into four progress indicators: ambition of targets, robustness of published data, implementation within company processes and direct substantive impact of company actions. Only emissions and energy flows within Scope 1 + 2 of the Greenhouse Gas Protocol[33] are evaluated.

reports[21,22]. International initiatives can enable decarbonisation in vulnerable regions and countries with developing economies[52]. Higher participation outside of the OECD in the SBTi and RE100 would indicate that these initiatives are achieving a supporting role in this, but such trends have yet to manifest. Future research should focus on whether or not the participation of companies in energy intensive sectors increases, if the initiatives themselves actually drive behavioural change or only legitimise ongoing trends, and how a more internationally diverse membership can be incentivised. The level of convergence and transparency between initiatives should also be studied to decrease the amount of data fragmentation, which remains a crucial barrier that the public and researchers must overcome for proper analysis.

## Methods
### Logical framework used in the study
We developed a Logical Framework[59,60] to evaluate the two initiatives featured in this study by adapting the work of ref. 11. This framework evaluates the steps taken by companies in climate initiatives sequentially: from setting appropriate targets, to improving their capacity and implementing changes, and finally evaluating the outcome of their actions. This is done through four indicators: Ambition, Robustness, Implementation and Substantive Progress. Although each indicator is composed of quantitative metrics, all four are qualitatively combined to evaluate and contextualise the mitigation potential of the initiatives. Table 2 describes the four indicators, as well as crucial concepts related to them.

Ambition is defined as the compatibility of company targets with trends put forward in IPCC reports[52]. In the case of the SBTi, targets aim at a certain percent of GHG emission reductions from a baseline year[25]. However, the initiative does not publish the baseline emissions of their members in absolute amounts in their website or reports[22] (as of the 23rd of February of 2021). To enable comparisons, we used the most recent CDP responses of members since they include a section for SBTi targets, including baseline emissions. Then, we compared these baseline emissions against historical emissions stated in prior questionnaires or reports. If a company submitted a baseline that exceeded historical trends, or did not disclose a baseline at all, it was assumed that the target covered 100% of the historical emissions of the baseline year. Then, we projected SBTi targets into the future by keeping the baseline emissions constant for years prior to the baseline year. For RE100, targets aim at increasing the share of renewable

electricity consumed by its members. To set target baselines, we used RE100 reports to obtain the renewable electricity ratios that companies disclosed to the initiative at the year of joining[21].

The scenarios used for comparison were taken from the IIASA database for the IPCC Special Report on Global Warming of 1.5 °C[37]. These were selected based on relevancy by ensuring they did not assume a global decrease in emissions between 2010–2020, which did not occur[58], that they were used in the SBTi's Absolute Contraction Approach[24,25], and that they were consistent with highly ambitious shared socioeconomic pathway narratives for achieving Paris goals[38]. The selected scenarios were SSP1-19 (consistent with 1.5 °C) and SSP1-26 (consistent with below 2 °C) produced by the AIM/CGE 2.0, GCAM 4.2 and WITCH-GLOBIOM 3.1 models.

Robustness relates to the actor's ability to achieve established goals, primarily influenced by the number of resources given to achieve the target[11]. However, in our case, we redefine it as the degree of trust that can be assigned to the environmental data published by companies to CDP and in self-published reports, given that this type of disclosure remains a voluntary exercise in most parts of the world[61,62].

First, by analysing the type of third-party verification employed by companies to validate their environmental data. Although CDP allows a variety of verification standards, they generally classify quality in five categories: None, Limited, Moderate, Reasonable and High[41]. Accounting literature typically agrees that assurance at a Limited and Moderate level applies to cases where there was "a reduction in work effort that would have otherwise been necessary to obtain more assurance"[44], with a lower level of certainty than their Reasonable and High equivalents[42,43]. Essentially, low assurance has less confidence and will employ a negative statement (e.g.,"we are not aware of any misstatements or modifications"), while Reasonable assurance will employ a positive statement (e.g., "the information seen in the report has been stated correctly")[31] (see the Supplementary Material for more information of the differences of verification qualifications).

Second, by analysing the visibility into their renewable energy purchases using Eq. (1):

$$\text{Visibility}_{c,i} = \frac{\text{PS2MB}^{RE}_{c,i}}{\text{PElec}^{RE}_{c,i} + \text{PHSC}^{RE}_{c,i}} \tag{1}$$

where Visibility is defined as the ratio between the renewable energy purchased ($\text{PS2MB}^{RE}$) with specific sourcing models disclosed in CDP questionnaires[46] or self-released reports and the total purchased

renewable electricity (PElec[RE]) and purchased renewable heat, steam and cooling (PHSC[RE]) disclosed by company $c$ in year $i$ in said questionnaires (e.g., section C8.2d[46]) or reports.

Implementation relates to the activities companies produced to achieve their targets, such as investing in renewable energy generation or switching to electric mobility. We analyse this indicator in two ways. First, by looking into the the energy profile of companies in terms of their use of renewable energy (RE) and non-renewable energy (NRE). We distinguish between internal consumption of fuel and self-generation of renewable electricity (RE Self-Gen), which relate to Scope 1 emissions, and energy purchases, which is attributed to Scope 2[31].

Second, by disaggregating the renewable energy purchased by these companies into four sourcing models of varying capacity to induce additional renewable energy generation[48]: Unbundled Energy Attribute Certificates (U-EACs), Utility Green Premiums (Utility GPs), Power Purchase Agreements (PPAs) and Self-Generated Renewable Electricity. U-EACs, also known as Renewable Energy Certificates or Guarantees of Origin, are a market-based instrument which is sold separately from energy products. They exist merely as a representation of the characteristics of the energy generated, and producers can choose to retire the credit themselves or to sell it to other parties[34]. Retiring a credit enables the owner to claim the $tCO_{2e}$ per MWh associated with it as their own, lowering their accounted emissions. They are considered of low additionality because they do not represent the grid-mix at the point of consumption[20] and because their low prices and short contractual periods do not offer enough reliability to investors[51,63]. Utility GPs are alternative products sold by energy suppliers that have an increased or full renewable energy content. Utilities can bundle them with Energy Attribute Certificates, inheriting some their issues, and in many cases the energy bought is produced by old renewable installations built through public support schemes[48,64], lowering their additionality. PPAs are long term arrangements between users and power producers which are generally seen as having a high potential to result in new renewable capacity installations[19,48]. Finally, Self-Generated Renewable Electricity is seen as the most additional model since it implies that the company invested in the capacity themselves[19].

Substantive Progress compares the targets analysed in the ambition indicator against actual collective Scope $1+2$ performance. For SBTi it relates to GHG emissions between 2015–2019, while for RE100 it is the overall share of renewable energy used in the same period.

## Data gathering

To correctly assess the companies featured in this study, we followed four steps when obtaining data: (1) creating an initial database of the G500 and classifying them by geographical region and energy sector, (2) identifying initiative members, (3) filling information gaps in the targets of each company, and (4) gathering environmental data from CDP and company reports. Databases of the Global 500, SBTi and RE100 members were generated on the same date, the 23rd of February 2021, by employing data scraping techniques. In the case of the SBTi, the initiative provides a downloadable Excel file with its members[65], so the scrapping stage was skipped.

Geographical region and energy sector values were obtained for all G500 companies using data scrapped from Fortune's website. Then, we classified companies in regional groupings by comparing head-quarters location data to Natural Earth databases[66] to obtain a World Bank Region classification for each of them. A list of classifications was also developed, based on sectoral nomenclature commonly used in policy research[53,67]. The list consisted of five sectors: Electricity Generation, Industry, Buildings (reclassified as Services), Transport and Fossil Fuel Production. The Industry sector was split further into Energy Intensive Industry (composed of businesses producing materials such as metals, glass, cement, and basic chemicals) and Light

Industry to represent differences in emissions intensity better[68,69]. Then, GICS[70] classifications were assigned to each company using data in market research websites. Finally, these GICS classifications were used to determine the energy sector most relevant to each company (see Supplementary Table 3 for an overview of how companies were reclassified).

Member identification was carried out by detecting overlaps between G500 companies and members listed in the initiatives' websites as of the 23rd of February of 2021 using approximate string matching techniques. This step was necessary since initiatives use different naming conventions and sometimes allow subsidiary companies to join, either alongside parent companies for SBTi[31] or under special conditions for RE100[27]. As a precautionary step, half of the resulting overlapping companies were subjected to randomised validation to ensure they were initiative members. Finally, the complete set was compared with preexisting reports[72] to identify possible omissions.

Target data had to be obtained differently for each initiative. For the SBTi, it was obtained from the initiative's website[65], and for RE100, the most recent progress report was used[21] (in the case of recent members, the RE100 website was used instead, although it tended to omit interim targets). At their most basic, targets consist of a baseline year, a baseline value (e.g., $tCO_{2e}$, renewable electricity ratio), a targeted reduction or increase, and a target year[24]. However, no initiative disclosed target data with complete transparency: baseline values were missing in the initiatives' websites at the time of the study, or they were given in percentages in annual reports[21,22], which made the impacts of the commitments indistinguishable between members. In the case of absolute targets in the SBTi (i.e., targets tracking only $tCO_{2e}$ emitted) and all RE100 targets, CDP questionnaires were used to establish baseline target values in more concrete terms (either $tCO_{2e}$ emitted in that year or total kWh of renewable electricity consumed). Data related to Intensity targets in the SBTi, which follow the Sectoral Decarbonisation Approach[26], was not collected because recent studies have highlighted that these targets suffer from additional transparency issues and often require data from unofficial sources[18] and this study focuses only on data officially disclosed by companies in our sample.

Environmental data collection involved constructing databases for 2015–2019 (see Supplementary Fig. 1). CDP questionnaires were preferred whenever available, although it was often necessary to review company annual reports and other sustainability documents to complete gaps or correct discrepancies. GHG emission collection followed the most recent GHG Protocol methodology[31,33,34]. Scope 2 (indirect) emissions presented unique problems, as the protocol allows two accounting methodologies for it: location-based (LB), where grid average emission factors are employed, and market-based (MB), which accounts for traceable energy certificates to adjust emission factors[73]. Although the GHG protocol states that LB disclosure is obligatory, companies often ignored this requirement and disclosed only MB values. In such cases, it was made sure that the Scope 2 data collected was of the same type as the emission target of the company (if applicable). Energy consumption and energy generation data were also obtained, following the latest CDP format[46]. In the case of Electricity Generation companies, energy consumption data was collected but not used for the study as it tended to be disclosed erroneously. The most recent questionnaires disallow utilities from disclosing energy consumption altogether, providing a sector-specific section for them instead. Data for renewable energy (RE) sourcing methods was also collected for energy end-use companies (i.e., MWh consumed through Power Purchase Agreements, Utility Green Premiums, Energy Attribute Certificates).

Not every company disclosed all types of environmental data. At a minimum, Scope 1, Scope 2 and energy consumption data had to be available for most years. If emissions or energy data were not disclosed in CDP questionnaires or public reports for a specific year, they were estimated during the data validation step. However, the companies

were discarded if they did not disclose sufficient data for more than 1 year. Nine companies had to be omitted due to such issues (see Supplementary Information).

## Environmental data validation

Once the data for a specific company was gathered, it was subjected to a series of validation tests to ensure it complied with the following criteria:

- Energy data is coherent and complete per year.
- Emissions data is consistent across years.
- Renewable sourcing data comply with GHG emission protocol guidelines per year.

These tests were done to diminish the effects of information issues identified by previous studies evaluating corporate environmental data disclosure through CDP[10,16,54]. Specifically, the influence of longitudinal changes in CDP's questionnaire, errors during submission, accounting errors in the firm's methodology and inconsistent reporting boundaries. Many of these issues stem from the fact that the energy consumption, self-generation, self-consumption, and low-carbon energy purchase sections in CDP questionnaires are not required to be mathematically consistent. This issue is exacerbated further in self-released company reports, which are much less standardised in their presentation. We designed our testing methodology to minimise these issues as much as possible, considering the black-box nature of company environmental disclosure.

Per-year tests involved correcting energy values disclosed by employing several equality and inequality tests. Any cases where the conditions did not hold were reviewed individually by comparing CDP data and public reports to catch issues such as magnitude errors, conversion errors, typing errors, empty values, and similar. The equality conditions in Eqs. (2)–(7) had to hold for each company ($c$) and year ($i$):

$$Energy^{T}_{c,i} = Energy^{NRE}_{c,i} + Energy^{RE}_{c,i} \qquad (2)$$

$$Energy^{RE}_{c,i} = Fuel^{RE}_{c,i} + PElec^{RE}_{c,i} + PHSC^{RE}_{c,i} + SGNonFuel^{RE}_{c,i} \qquad (3)$$

$$Energy^{NRE}_{c,i} = Fuel^{NRE}_{c,i} + PElec^{NRE}_{c,i} + PHSC^{NRE}_{c,i} \qquad (4)$$

$$Fuel^{T}_{c,i} = Fuel^{NRE}_{c,i} + Fuel^{RE}_{c,i} \qquad (5)$$

$$PElec^{T}_{c,i} = PElec^{NRE}_{c,i} + PElec^{RE}_{c,i} \qquad (6)$$

$$PHSC^{T}_{c,i} = PHSC^{NRE}_{c,i} + PHSC^{RE}_{c,i} \qquad (7)$$

where $Energy^{T}_{c,i}$, $Energy^{RE}_{c,i}$ and $Energy^{NRE}_{c,i}$ are the annualised totals of all energy, renewable energy and non-renewable energy consumed, respectively. Each is made up of several sub-categories present in CDP questionnaires[32,46]:

- Fuel: fuel consumed within the company's organisational boundary.
- PElec: consumption of purchased electricity produced outside the company's organisational boundary.
- PHSC: consumption of purchased heat, steam and cooling (HSC) produced outside the company's organisational boundary.
- SGNonFuel$_{RE}$: consumed self-generated non-fuel renewable energy (e.g., solar, wind, geothermal) that is owned and operated by the company.

The inequality tests in Eqs. (8)–(10) were employed for each company ($c$) and year ($i$) to verify their disclosed energy generation data:

$$GGElec^{T}_{c,i} \geq GGElec^{RE}_{c,i} \qquad GGHSC^{T}_{c,i} \geq GGHSC^{RE}_{c,i} \qquad (8)$$

$$GGElec^{T}_{c,i} \geq SCElec^{T}_{c,i} \qquad GGHSC^{T}_{c,i} \geq SCHSC^{T}_{c,i} \qquad (9)$$

$$GGElec^{RE}_{c,i} \geq SCElec^{RE}_{c,i} \qquad GGHSC^{RE}_{c,i} \geq SCHSC^{RE}_{c,i} \qquad (10)$$

where GGElec and GGHSC represent gross generation of electricity and HSC, respectively. Similarly, SCElec and SCHSC represent self-consumed electricity and HSC. Since CDP questionnaires prior to 2017 did not include HSC generation data; values for these years were assumed to be zero if the company had no HSC generation in 2017. Otherwise, the data was sourced from self-released reports if available.

Cross-year consistency tests involved comparing energy consumption trends against emission trends in Scopes 1 and 2 (LB, MB). Disparities (such as emission decreases without apparent changes in energy trends) were reviewed by reviewing company emission reports. If a problem was identified (such as inconsistent company boundaries, pervasive methodology errors or conversion discrepancies), corrected data was taken from later CDP responses or self-released reports. In cases were this was not possible, emission factors ($FS1_{c,i}$, $FS2LB_{c,i}$, $FS2MB_{c,i}$) of adjacent years were averaged in order to estimate the affected year using Eqs. (11)–(13):

$$FS1_{c,i} = \frac{S1_{c,i}}{Fuel^{NRE}_{c,i}} \qquad (11)$$

$$FS2LB_{c,i} = \frac{S2LB_{c,i}}{PElec^{T}_{c,i} + PHSC^{T}_{c,i}} \qquad (12)$$

$$FS2MB_{c,i} = \frac{S2MB_{c,i}}{PElec^{NRE}_{c,i} + PHSC^{NRE}_{c,i}} \qquad (13)$$

where S1, S2LB, and S2MB are the reported Scope 1, Scope 2 Location-based and Scope 2 Market-based emissions of the years adjacent to the affected value, in $tCO_{2e}$. As a final step in this stage, the emission factors for each company were normalised with their earliest available value, and a detection test was applied to identify cases where factors increased 100% over the norm (see Supplementary Fig. 2).

Only energy end-use companies could be tested thoroughly. Our tests could not apply to companies in the Electricity Generation sector due to unique disclosure issues. First, CDP questionnaires for 2015–2016 did not provide crucial data such as gross/net generation, installed capacity or per-technology emissions. Second, Scope 2 emission calculations have additional complexities due to transmission losses being included within them if the company also has a transmission or distribution business[34]. Lastly, self-released reports by these companies did not provide enough detail to construct the energy consumption section. Instead, the emissions of these companies were compared against the amounts stated in sustainability reports to identify errors during CDP submission. A detailed document for this sector is available in the database, explicitly stating the reviewed documents as well as issues that were identified and corrected.

Renewable sourcing model tests evaluated if the data disclosed in the Scope 2 Market-based section of CDP responses violated GHG protocol guidelines[34]. When available, text entries and emission factors that the companies included in CDP responses were used to identify if they included grid-mix renewable energy or non-renewable energy sources in the section. Similarly, the amounts disclosed were compared to renewable self-generation data provided in other sections or

reports to identify if the company counted contractual instruments such as PPAs as their own. Nuclear energy was also removed if identified as it is not considered valid under RE100 criteria. Due to constant year-on-year changes to the names used by CDP for this type of data, it was common to see companies refusing to use the updated terms, necessitating further corrections (see Supplementary Table 2). Once these corrections were applied, a final per-year test was run to ensure the reported renewable sourcing did not exceed actual consumed purchased renewable energy using Eq. (14):

$$\text{PElec}_{c,i}^{RE} + \text{PHSC}_{c,i}^{RE} \geq \text{PS2MB}_{c,i}^{RE} \tag{14}$$

where $\text{PS2MB}^{RE}$ is the total renewable energy sourced through valid purchase instruments by company $c$ in year $i$.

## Limitations

This study has limitations in its methodology and scope, and it is paramount to account for them when interpreting its results.

First, although the study focused on correcting disclosed data for longitudinal consistency, it has a limited capacity to detect cases where companies systematically misreport environmental data (intentionally or by misunderstanding). Other recent evaluations have showcased issues in how companies calculate and disclose their emissions to platforms such as CDP[19,54]. Similarly, third-party verification providers may also fail to perform their due diligence, leading to a further decrease in data quality[74,75].

Second is the omission of Scope 3 emissions, which in some cases can account for more than half of the total emissions within the control of these companies. This decision was necessary, as the level and quality of the disclosed Scope 3 categories varied widely between members. In many cases companies did not subject all their published Scope 3 categories to third-party verification, leaving no benchmark to evaluate data quality. This scope is often the least successfully mitigated[18], and this should be kept in mind when interpreting results. Nevertheless, we consider an analysis of just Scope 1 and 2 emissions to be meaningful because these emissions can be most directly controlled by companies and are often the first to be tackled in business GHG action plans.

Third, our analysis was limited to just two initiatives and covered only the business aspect of transnational climate initiatives. Other types of actors, such as cities, regions and forest initiatives[2,4] are exempt from it and may bring emission reductions on their own accord.

Fourth, our study could not identify if involvement in these initiatives triggers higher climate ambition in companies or if membership is a by-product of already strong commitment towards it. The chief reason for this was a lack of comparative analysis between participating companies and similar entities not actively involved in this type of initiative. Similarly, data was not collected for years before the initiatives' creation.

Fifth, there are limitations on the comparability of company emissions under the GHG Protocol. It is possible that overlaps exist between Electricity Generation and other sectors, reducing mitigation effectiveness, but publicly available information was insufficient to identify such cases since there is no requirement for companies to release it. Similarly, the GHG Protocol itself is being subjected to scrutiny due limitations in its capacity to differentiate operational changes from accounting choices[76]. A more targeted study, perhaps evaluating the contractual preferences of high-emitting participants and their accounting methodologies, would aid in addressing these gaps.

Finally, our study only quantifies some aspects of robustness and implementation. Other possible metrics include appointed staff, certifications obtained, knowledge production, training, lobbying and participation in conferences and other events[3]. These aspects may be critical to contrast individual success or failure cases between these firms, and qualitative analyses of them might prove to be a more effective tool for their evaluation.

### Reporting summary
Further information on research design is available in the Nature Portfolio Reporting Summary linked to this article.

## Data availability
The evaluated company data generated in this study have been deposited in the 4TU.ResearchData repository[77] under accession code access. The raw CDP questionnaire data are available under restricted access due to limits set by CDP, access can be obtained by registering in the CDP website at https://www.cdp.net/. The scenario data used in the study are available in the IAMC 1.5° Scenario Explorer at https://data.ene.iiasa.ac.at/iamc-1.5c-explorer/.

## Code availability
All scripts used to collect company and initiative data, as well as to produce the results featured in this study, can be accessed in the 4TU.ResearchData repository[77] under accession code https://doi.org/10.4121/16616965 and at https://github.com/irm-codebase/Quantitative_evaluation_SBTi_and_RE100.

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

## Acknowledgements

This research was supported by a 2019 scholarship awarded to I.R.M. by the Mexican National Council of Science and Technology (CONACYT) and the Council of Science and Technology of the State of Jalisco (COECYTJAL) under reference 2019–000026–01EXTF–00053. We thank M. A. Eltahir Elabbas and M. Šalandová for their feedback and inputs on the work.

## Author contributions

I.R.M. and K.B. designed the concept and focus of the work. I.R.M. collected and analysed the data, designed the software and validation infrastructure, and drafted the paper. K.B. supervised the work and co-wrote the paper.

## Competing interests

The authors declare no competing interests. The author K.B. was appointed to the Chair of the SBTi Technical Council after the review process of this article.
