## [Peer Review File · Nature Communications]

Quantitative evaluation of large corporate climate action initiatives shows mixed progress in their first half-decadeREVIEWER COMMENTS

Reviewer #1 (Remarks to the Author):

This paper covers an issue of high contemporary relevance, namely investigating evidence of the implementation of voluntary climate initiatives and corresponding targets, the quality of the efforts made, and their additionality. The SBTi and RE100 are prominent examples of these initiatives and have been subject to criticism on both the validity of their targets and the implementation of them. The paper claims to show that the largest 102 companies by revenue across both initiatives have reduced their total emissions by 35% between 2015 and 2019, with electrification a component of this. It warns that targets to 2030 are consistent with IPCC pathways for below 2C, and electricity sector targets, for 1.5C, but that the steps to be taken to reach this are not clear and rely on third-party verification and poor quality renewable energy purchases.

Based on a close reading, I would recommend substantial revisions to content and framing.

General comments

- The clarity of the writing could be improved as it can be quite hard to follow in places. This is not a comment on the content, just on the writing style. There are in the introductory sections a few places where terms are not defined or explained and remain quite inaccessible to readers not already familiar with the initiatives being evaluated.
- The focus on robustness of target-meeting disclosure is very helpful and may even deserve greater prominence, as it is the litmus test for whether ambition is just a marketing exercise, or has real substance. It may be worth reporting in the abstract some of the statistics on the distribution of verification quality - with the majority being classified as limited, and very little high-quality. If there is evidence available, it may be helpful to provide a little context on what makes these verifiers' quality limited and the extent to which conflicts of interest and insufficient due diligence are a problem in the disclosure industry.
- There is ambiguity over what is meant by "additionality" in this paper. The authors seem to use this term with reference to the instruments used to procure renewable energy. What does not seem to be addressed but should be even if the data is insufficient to provide much insight, is the extent to which companies' actions to decarbonise themselves are "additional" to what would have happened without these targets. The trends, which seem to indicate some companies' emissions falling prior to setting targets, suggest perhaps not. Since this paper is concerned with companies' success in meeting their own targets, it would be very relevant to discuss with more focus how much active change has been required for these companies to achieve them. Similarly, it would be helpful to see more critical engagement with the possibility that declining energy intensity/changing composition of business activities/declining carbon intensity of electricity have allowed companies to claim responsibility for the observed reductions where in fact they would likely have occurred anyway due to wider trends in, for example, the cost of renewable energy, the rules governing dispatch of electricity in power markets, etc. The lack of clarity over this issue means claims that the observed trends in the data "demonstrate that the target setting approaches set by each initiative are sound" (p.8) are unconvincing.
- Excluding Scope 3 emissions is a major omission since it comprises the bulk of many large companies' emissions. While this omission is understandable due to data limitations, the authors should be more upfront about this in the abstract and introduction, noting that their results are confined to direct emissions and emissions from energy purchases and exploring the implications of only looking at Scope 1 and 2 in more detail, rather than referring to this in the last sentence of the paper without any analysis of how this limits or qualifies the findings in the paper.
- The claim in the discussion section "that the target setting approaches set by each initiative are sound" does not necessarily follow from the results (see general comments). Why does the fact of emissions reductions ahead of these targets demonstrate that the approach is sound? Please consider re-wording less strongly and acknowledging more clearly that there are reasons why the targets have been met that may be unrelated to the initiatives and the target setting approach itself. This is a

subtle but very important nuance as this sort of phrasing can give the misleading impression that the initiatives have not only successfully corralled companies into setting targets but that this process was the only relevant factor in achieving them, which the authors have not shown convincingly. And as the discussion goes on to say on p.9, the demonstrated reductions have taken place among a handful of companies, with little evidence that the target-setting exercise has achieved anything much beyond increased renewable energy procurement, which in any case may well have happened irrespective of target-setting (see general comments on additionality). This finding, i.e. that the vast majority of emissions reductions in this analysis are due to renewable energy generation and procurement either directly, or more often through instruments of varying quality, and very little fundamental (Scope 1) or cross-value-chain (Scope 3) progress has been made on emissions, could be made more prominent as a central result of this analysis that would also make clearer the limitations of the data and climate actions taken.

Specific comments

p.1

- "Qualitative assessments have shown other problems that exacerbate data issues, such as lack of permanent staff, adequate funding or measurable goals". Does this refer to staff/funding/goals of voluntary platforms, or to resources committed voluntary initiatives within companies? May be worth clarifying and/or expanding.

p.2

- Regarding the NCI and SBTi studies, it is also worth noting that there is not necessarily any evaluation in those studies (unless the authors know otherwise) of whether the targets are appropriately ambitious, which adds to the problems of insufficient transparency and the widespread use of poor-quality offsets and renewables instruments.

- The statement on the Lui et al study is missing key pieces of information required to assess its validity. Suggest remarking on how much of an issue ignoring overlaps (i.e. double counting) is, and how large each of the hypothetical 2000 members would have to be in terms of emissions footprint for their estimates to be met.

- Please provide more information, either here or elsewhere in the paper, on how valid the qualification process is, and how robust SBTi's method for assessing consistency with 1.5C/2C pathways is.

- It is not clear what the "two year waiting period" in the SBTi contact refers to. Please clarify.

p.3

- Authors refer to "financial businesses with low emissions" in RE100. Does this include financed emissions in their portfolios?

- Please explain why fossil fuel producers can commit to SBTi targets but they cannot be approved. Is it because SBTi does not engage with these companies or is it a question of measurement?

p.4

- First sentence is long and grammatically confusing - please revise. Perhaps "falling outside...2015-2019" should be placed in parentheses and semicolons used to separate list items. Please also explain more clearly why 2019 is the cutoff point.

- What does "evaluated partially" mean with reference to intensity targets, and why do they suffer more from "disclosure problems" than absolute targets?

- Why are emissions held constant for years before the baseline, if not a lack of data? This might give the misleading impression of constant historical emissions for a company for whom emissions were already falling for other reasons (as is pointed out on p. 7).

- Please explain in more detail, or refer to further methodological documents, what you mean by comparing company-level trends to "normalized scenarios for OECD+EU nations". Does this mean that the additionality of emissions reductions is being calculated against economy-wide emissions trends? This is probably not a fair comparison as the companies represented in the same may tend towards

less emissions-intensive activities than the wider economy although it appears that your assessment of consistency with SSP1 is based on sector-specific pathways. If this interpretation is not correct, please clarify. Please also extend this clarification to explaining how the baseline figure is calculated - is this done consistently across all companies on the basis of a single baseline year?

p.5

- Figure 3 - the right hand panel (b) doesn't seem very necessary or additional to (a). Suggest keeping just (a).

p.6

- Why did CDP bar financial companies from disclosing RE purchasing methods?

p.7

- Please explain more clearly why U-EACs are considered lower quality than PPAs and why their displacement of FF sources and emissions reductions attributes are different to those of PPAs. The logic is not made explicit. If the logic is indeed sound, the authors should draw attention to most of the RE procurement being of this lower quality, but also consider why some companies might choose U-EACs over PPAs: are they cheaper, for example? If so, why?

- In measuring the "rate of consumption of high additionality instruments per member" you should really be normalising against revenue or total electricity consumption, since without this, variance in total RE purchased between RE100 and SBTi could just be explained by differences in company size/emissions intensity (and perhaps it isn't but we can't tell from the data provided)

- Please reconsider the word choices "good trends" and "impressive" in relation to AAGRs for SBTi members with intensity targets. These are subjective assessments and should be treated more objectively in this paper.

- Please clarify whether the disparity between RE100 reports and public disclosure indicates an underreporting or overreporting of RE consumption. Currently the authors use the word "offset", which is ambiguous.

p.8

- It is not clear from the results whether or not Scope 1 electricity sector emissions reductions are being double counted in the form of Scope 2 emissions reductions in energy intensive industry or other sectors that consume this electricity.

p.9

- Spelling error: "where utility companies" should be "were utility companies"

- Suggest qualifying the claim that "companies have successfully met the goals established by these two initiatives". This study does not show this; it shows that a subset of disclosing firms (who, by disclosing their emissions, is probably biased towards those more likely to be achieving their targets) are on trend to meet targets that are for 2030 (RE100) and targets that are set and verified by the companies themselves, not directly (SBTi).

Reviewer #2 (Remarks to the Author):

I enjoyed the material presented in the paper and believe, on the basis, of the analysis that the materials accurately reflect what is transpiring in the initiatives you have tracked. There are, however, matters of framing that are likely to fundamentally reshape how you might understand the data you present. Both of these comments require a more nuanced view of corporate functioning, carbon accounting and accountability. It is perhaps an omission to have conducted this work without the obvious input of a specialist in this area. The two issues of framing then ...

1. You state in the opening of the paper that contributions by corporations to emissions reductions will be additional to NDCs. This is not likely to be true because the choices that corporations make for

reductions are set within the context of country actions to employ policy levers to reduce emissions. Many of these emission reductions, then, will be counted by both countries and corporations. At the end of the day, it is the country reduction trajectories that matter with the picture of corporate reductions providing a different 'cut' through much of the same activities. To spell this out a little further. If a country has an emissions reduction programme in place that requires companies (as the subject of regulation) to reduce their emissions then your rationale would count these reductions twice – when in fact they are one in the same reductions, just with different entities articulating their timing and nature.

2. At the top of page 5 you note that the reductions 'exceeds the ratio for OECD + EU nationals where most of these companies have their headquarters'. This seemed a very odd comparison to make. The companies you have in the same are large and therefore likely to have operations in many different countries. Their emissions totals, therefore, do not necessarily have any connection to their headquartered countries because they will be the sum of emissions in a number of countries. This makes comparing reduction ratios non-sensical.

Alongside these framing issues there are other aspects that could usefully be addressed:

- While I can appreciate why scope three was outside of the analysis (at least because of the inconsistent way in which such measures are made), scope 3 emissions for any individual company are frequently many factors greater than scope 1 and 2 emissions. Some way of acknowledging this fact would be helpful and what it means for the thrust of your argument are non-trivial.
- At page 3 you note that Chinese companies stood out as absent – which is not surprising. While their companies may be very large, they often have a significant percentage of ownership with the Chinese state. Where this is the case, their propensity to participate in voluntary corporate initiatives will be limited. This is a common observation in the accounting/business/management literature – that Chinese corporations work within very different sets of norms and (more generally) that corporate ownership profiles generate different propensities for pro-environmental behaviours and reporting practices.
- There is a need to better understand why limited assurance is likely for voluntarily disclosed carbon information. The issue here is one of costs and benefits as well as the complexity of the greater level of data certainty required for what is often unregulated measurement protocols (including estimations from emission factors cf direct measurements). Moreover, the highly distributed data sets that will usually sit behind Fortune 500 company reports make limited assurance the most suitable form. It is notable in your data that those sectors (eg utility companies in countries with high levels of regulatory oversight) are able to subject to reasonable assurance. This is a function of underlying data complexity and capture.
- You note that there are year by year inconsistencies in CDP data but I am not sure what you mean by this. The standards and approaches are not likely to be consistent as measurement protocols (specified by countries) can change, emission factors change as energy systems are transformed, figures may vary depending on changes in corporate activities and the CDP itself changes its information needs. This co-evolution of these factors will mean that consistency is probably not available as the approaches reflect underlying changes.
- You note that membership of the initiatives is concentrated on OECD countries – which is a reflection on your data being based on the Global 500 list. This observation is driven by your sample design and hence its relevance to a line of argument is uncertain.
- In places you talk about improved, centralized and converging disclosure practices ... what might be of more relevance is measurement protocols on which disclosure is based. This confusion is also obvious at the topic of page 2 when you say that the GRI leaves plenty of leeway for how companies

account (ie measure) and publish (ie reporting) environmental data. It is worth noting that, most usually, the WRIs GHG Protocol is the measurement tool (unless country regulations stipulate a different measurement) while the GRI is a reporting standard.

- On page 9 in the second to bottom paragraph you refer to studies [11, 15, 18, 48]. I can't locate reference 48 in the list and 11 and 15 are not obviously references for the point you are making.

Reviewer #3 (Remarks to the Author):

Key results

The manuscript investigates ex-post the effects of two important climate initiatives, SBTi and RE100, and the progress of their participating organisations on Scope 1 and Scope 2 climate actions, from 2015 to 2019. Notably, the study highlights that collectively all analysed SBTi organisations are aligned with IPCC pathways for well below 2°C, whereas REC100 organisations show a share of renewable electricity consistent with well beyond 1.5 °C requirements. Surprisingly, organisations have indeed reduced their Scope 1 and Scope 2 CO₂ eq. emissions by 35% compared to the starting baseline. However, the bulk of this result was achieved by a small group of five utilities and energy-intensive companies, accounting for 85% of the total reduction. Therefore, the study shows mixed results once a more in depth-analysis is carried out and this is extremely interesting as it sheds lights on some numbers that otherwise would be misinterpreted. The same goes for renewable energy adoption. Although the share of renewable purchasing is increasing, the quality of purchasing instruments do vary. In addition, the study provides further insights compared with previous literature by assessing organisations on their ambition, methodology, implementation, and progress on climate actions.

Overall, the study offers updated and useful data and information to evaluate and monitor the validity of those two relevant on-going climate initiatives on scope 1 and scope 2 emissions, also assessing organisations progresses in reaching science-based targets.

Validity

Data interpretation and conclusions are sound. Strong evidence is provided for the authors' claims and all appropriate controls have been included. However, the paper missed a data validation test on energy intensive and utilities organisations.

Significance

The present work is extremely interesting for both the literature and practitioners.

On the one hand, the few ex-post studies on the literature focus on the achievement of SBTi targets by mixed small and large companies, but they do not focus and express the total impact (quantitative data) of large organisations. Differently from previous literature that expresses a certain percentage of analysed organisations to be on track with SBT and IPCC requirements, this study evaluates altogether the efficacy of both initiatives to match such requirements by studying the total amount of CO₂ emission reduction and the increase of renewable energy usage of the analysed large organisations. It provides also further insights than the previous literature due to the evaluation of the robustness of the verification processes.

On the other hand, ex-post data analysis is extremely useful for practitioners as they can be used as monitoring tools but also to strengthen the initiatives themselves by focusing on the improvement needs and limits for monitoring highlighted by the paper.

However, the exclusion of scope 3, considering the sample of the study, might limit the relevance of this paper.

Data and methodology

Data quality, methodology and quality of presentation are sound for the aims of the article. Details are properly provided for reproducibility; however single company excels could be useful to check on database quality.

The analytical approach properly follows the phases of data gathering, environmental data validation, benchmarks and progress. All the phases are thoughtfully designed for the objective of the paper and to increase data quality as much as possible considering the intrinsic data problems. Supplementary materials provide understanding of the methodology and some results. A weblink for a public database with all datasets is fully available.

5.1 Data gathering

Data gathering approach and the selection of databases are appropriate. Geographical region and energy sector values were associated with GICS categories for worldwide recognition. Members identification was properly carried out avoiding naming issues and the complete set was further assessed for omissions.

Target data were obtained from the most recent sources. The usage of SBTi data only under the absolute contraction approach instead of sectoral decarbonisation ones is justified. CDP questionnaires were appropriate and relevant sources to collect data in order establish baseline target values in more concrete terms.

The whole methodology followed to gather environmental data for all SBTi and RE100 participants in the G500 is logic and sound, and clearly visible in supplementary materials. Database construction is made with CDP questionnaires, annual reports and sustainability documents to be able to obtain all relevant data since data lacking is one of the most important problems in environmental disclosure. The approach is appropriate. However, from the data sets it is not clear which data are obtained from which documents. Moreover, authors say that whenever possible, "Scope 2 data collected were of the same type as the emission target of the company". Since the use of different emission factors between market-based and location-based might be extremely relevant for the count of the total amount of GHG emissions, it could be relevant to have an understanding of the percentage of location and market based used for the calculations. In fact, SBTi companies can choose either one or another approach to set scope 2 targets. Consequently, it is not clear in the text and figure 2 which data (LB or MB) have been prominently used.

5.2 Environmental validation

The environmental validation data design is well structured to diminish the effects of information issues. The methodology designed by the authors is robust. However, considering that the highest emissions, as found out by the authors, are from the utilities sector, it is a pity that "Our tests were not applied to companies in the Electricity Generation sector due to their unique nature". Probably, some tests to diminish the effects of information issues on such data would have been relevant. Per year tests, cross-year consistency tests and renewable sourcing tests are robust and innovative methodologies. Considering the variety of emissions factors used across years by the same organisation as well as among different organisations, equation 10,11 and 12, with figure 8, are extremely interesting and very much appreciated methods to alleviate the differences.

5.3 Benchmark and progress indicators

The logical framework employed in the literature is consistent and the author's adaptation clear in the supplementary material. The description and considerations of the methodology are appropriate.

Suggested improvements

Although data sets are available and usable for statistical analysis, I kindly ask for the coding scripts. I also ask for the individual files for each company, to further check on the accuracy of data collection. The manuscript is really sound and valuable. However, my main concern is about the relevance of this research. The authors used a sample mainly based on services and light industry and they considered

only scope 1 and scope 2 because as they rightly pointed out there are no data for scope 3. They have done a very good job with the data available but not to consider scope 3 impacts for such industries is a relevant shortcoming since their impacts are mainly in that scope. I fully know that this shortcoming cannot be solved easily, but it had to be pointed out.

Nonetheless, it is possible to suggest some precise improvements to work on:

- It would be suggested to evaluate whether the following recent paper Bjørn, A., Tilsted, J.P., Addas, A. et al. Can Science-Based Targets Make the Private Sector Paris-Aligned? A Review of the Emerging Evidence. *Curr Clim Change Rep* 8, 53–69 (2022) could be useful to the introduction and discussion.
- It would be suggested to further exploit figure 6 by inserting other graphs showing the differences in progress between utilities and/or energy intensive companies and the other organisations. In fact, it is extremely relevant, as the authors point out in the discussion, that 85% of all GHG emissions reduction comes from a few energy intensive organisations. To enhance the clarity of figure 6 as well as the relevancy of the manuscript, it is then suggested to make graphically visible this important result after the overall graphs already present in figure 6;
- It would be suggested to make clear which data are taken from CDP, and which from other sustainability documents when CDP data were lacking. This strengthens the accuracy of environmental data collection and assures reproducibility for future studies. I asked for the individual files to see whether these specifications are present. There could be a table in supplementary material or in one file excel where there is the percentage of data taken by CDP and by the other documents.
- Since the authors assess only scope 1 and scope 2, it would be appropriate to insert in the supplementary material the total amount and percentages of scope 2 GHG emissions under MB and LB approaches that are aligned with companies' targets, as well as the contribution of MB and LB data to the total GHG amount (figure 2 and figure 6). It would not change the results, but it would give higher clarity and reproducibility.
- Following the previous suggestion, it would be suggested to make clearer in the text which data between MB and LB are the most prominent in the calculation (even though indirectly it should be MB).
- It would be suggested, if possible, to try to perform a validation test (a different one, if possible) also on electricity generation sector companies, to cover the whole sample and guarantees further robustness to data collection.

Clarity and context

The manuscript is clear in all its forms: manuscript's structure, content and data interpretation, methodology, language, figures and tables. The methodology and results have been provided with sufficient context and consideration of previous work. A few suggestions are already detailed in the previous section.

We thank the authors and the editor for the reporting summary.

References

The manuscript adequately builds on the previous literature. References are up to date. When references are not recent, they are nonetheless extremely relevant for the paper.

Author Responses to Initial Comments:

Reviewer #1 (Remarks to the Author):

This paper covers an issue of high contemporary relevance, namely investigating evidence of the implementation of voluntary climate initiatives and corresponding targets, the quality of the efforts made, and their additionality. The SBTi and RE100 are prominent examples of these initiatives and have been subject to criticism on both the validity of their targets and the implementation of them. The paper claims to show that the largest 102 companies by revenue across both initiatives have reduced their total emissions by 35% between 2015 and 2019, with electrification a component of this. It warns that targets to 2030 are consistent with IPCC pathways for below 2C, and electricity sector targets, for 1.5C, but that the steps to be taken to reach this are not clear and rely on third-party verification and poor quality renewable energy purchases.

Based on a close reading, I would recommend substantial revisions to content and framing.

We thank the reviewer for the suggestions, feedback and acknowledgement of the relevance of this work. A point-by-point recount of the changes can be found below.

General comments

- The clarity of the writing could be improved as it can be quite hard to follow in places. This is not a comment on the content, just on the writing style. There are in the introductory sections a few places where terms are not defined or explained and remain quite inaccessible to readers not already familiar with the initiatives being evaluated.

To address this concern we have made the logical framework used in the study more explicit by including it in the methods, as well as explanations for emissions scopes, the renewable sourcing methods and other terminology. We hope that this aids in making the study easier to follow for readers unfamiliar with corporate emissions accounting and climate initiatives.

Introduction:

“We evaluate the SBTi and RE100 initiatives in three steps. First, we identify their largest members by revenue using the Fortune Global 500 (G500) list of 2020 [27] by employing approximate string matching techniques, identifying the location of their headquarters and classifying them in sectors by their use of energy. Second, by creating a database with their emissions, energy use, energy purchasing preferences and climate targets using CDP responses and self-released environmental reports, validating the data to reduce errors in their disclosure. Finally, we employ the logical framework developed by Hale et al. [10] to assess climate action, descriptively evaluating these initiatives against metrics relevant to their goals in terms of the ambition of their targets, the robustness of their disclosure practices, how they implement changes in terms of energy usage and the substantive impact these practices have had on their emissions and use of renewable energy (see Methods). The study only concerns emissions accounted as Scope 1 and Scope 2, which are direct emissions produced within company boundaries and indirect emissions from energy purchases, respectively [28]. We do not evaluate targets related to Scope 3, which accounts for emissions from upstream and downstream activities [29].”

Methods:

Robustness:

“...although CDP allows a variety of verification standards, they generally classify quality in five categories: None, Limited, Moderate, Reasonable and High [38]. Accounting literature typically agrees that assurance at a Limited level applies to cases where there was “a

reduction in work effort that would have otherwise been necessary to obtain more assurance” [39], with a significantly lower level of certainty than a Reasonable equivalent [40, 41]. Essentially, Limited assurance has less confidence and will employ a negative statement (e.g., “we are not aware of any misstatements or modifications”), while Reasonable assurance will employ a positive statement (e.g., “the information seen in the report has been stated correctly”) [29].

Implementation:

“..four common sourcing methods with different degrees of potential additionality [45]. **Unbundled Energy Attribute Certificates (U-EACs, also known as Renewable Energy Certificates)** are decoupled from energy contracts [33], with producers deciding if they’d rather retire the credit themselves (thus lowering their emissions), or sell it to another party who can claim it as their own. They are considered to be of low additinality due to difficulties demonstrating their impact on new renewable installations [61], low prices [45] and short contractual periods [46]. Utility green products (Utility GPs) are sold by energy producers at a premium, and may come from installations built through public support schemes [45]. Power Purchase Agreements (PPAs) are long term arrangements between users and power producers which are generally seen as having a high potential to result in new renewable capacity installations [19, 45]. Finally, self-generated renewable electricity is seen as the most additional since it implies that the company invested in the capacity themselves [19].”

Summary table:

Table 2: Logical Framework used to evaluate the SBTi and RE100 in this study.

Adapted from¹⁰. It is subdivided into four **progress indicators**: ambition of targets, robustness of published data, implementation within company processes and direct substantive impact of company actions. Only emissions and energy flows within Scope 1+2 of the Greenhouse Gas Protocol²⁸ are considered.

Type of progress	Benchmarks and baselines	Key indicators	Period
Ambition	Extended company targets (Scope 1+2) SSP1-19 scenarios (normalised) SSP1-26 scenarios (normalised)	Targets within 1.5°C or 2°C trends	2015-2030
Robustness	Third-party verification qualification Claimed Renewable Energy purchases Transparent Renewable Energy purchases	Share of Reasonable to High verification High visibility into sourcing preferences	2015-2019
Implementation	Non-Renewable Energy used or produced Renewable Energy used or produced Renewable Energy sourcing model employed	Renewable Energy shares Share of high additionality sourcing	2015-2019
Substantive (direct impact)	Extended company targets (Scope 1+2) Actual collective GHG emissions Actual collective Renewable Electricity use	Performance against targets Change in GHG emitted Change in renewable electricity use	2015-2019
Causal impact	Overall effectiveness	All the above	Ex-post review

- The focus on robustness of target-meeting disclosure is very helpful and may even deserve greater prominence, as it is the litmus test for whether ambition is just a marketing exercise, or has real substance. It may be worth reporting in the abstract some of the statistics on the distribution of verification quality - with the majority being classified as limited, and very little high-quality. If there is evidence available, it may be helpful to provide a little context on what makes these verifiers' quality limited and the extent to which conflicts of interest and insufficient due diligence are a problem in the disclosure industry.

We thank the reviewer for their comment.

We now address the topic of Robustness in the **abstract**:

“We highlight how intermediate steps regarding data robustness and implementation of sustainability measures are lacking: **75% of public company data has only limited independent**

verification, and 71% of renewable electricity is sourced through low-impact or unknown sourcing mechanisms.”

We also detail the differences between Limited and Reasonable verification in the **Methods** section: “...although CDP allows a variety of verification standards, they generally classify quality in five categories: None, Limited, Moderate, Reasonable and High [38]. Accounting literature typically agrees that assurance at a Limited level applies to cases where there was “a reduction in work effort that would have otherwise been necessary to obtain more assurance” [39], with a significantly lower level of certainty than a Reasonable equivalent [40, 41]. Essentially, Limited assurance has less confidence and will employ a negative statement (e.g., “we are not aware of any misstatements or modifications”), while Reasonable assurance will employ a positive statement (e.g., “the information seen in the report has been stated correctly”) [29].”

Disclosure industry:

Addressing possible conflicts of interest and lack of due diligence in the disclosure industry would necessitate an entirely different type of study and methodology. Still, we believe that our findings should be communicated in order to aid such research.

- There is ambiguity over what is meant by "additionality" in this paper. The authors seem to use this term with reference to the instruments used to procure renewable energy. What does not seem to be addressed but should be even if the data is insufficient to provide much insight, is the extent to which companies' actions to decarbonise themselves are "additional" to what would have happened without these targets. The trends, which seem to indicate some companies' emissions falling prior to setting targets, suggest perhaps not. Since this paper is concerned with companies' success in meeting their own targets, it would be very relevant to discuss with more focus how much active change has been required for these companies to achieve them.

This request is now addressed in an improved **discussion** section, which provides comparisons against previous studies non-state action estimates by the UNEP, as well as a discussion on the complexities of accounting for actor additionality.

“These issues were forewarned by studies expressing worries over the additionality and quality of initiatives outside the UNFCCC [5, 6]. Also, the UNEP assessment of the impact of international cooperative initiatives for 2020 [48] already assumes that about one third of the emission reductions achieved would not be additional to national climate action. Given the fact that a substantial part of the actions implemented were through renewable energy sourcing, and this is for more than half done by using low-additionality sourcing methods, it can be expected that until 2019 the overlap with national climate action is higher than what was suggested in the UNEP report. Separating an initiative’s progress from national trends is not straight forward process since lack of action by other actors may counteract their benefits [4]. Determining the level of additionality more accurately would require in-dept investigation at the company level, or comparison participating and unengaged businesses in different sectors and regions, which are clearly important next research steps. Since our results reveal that mitigation progress is heavily concentrated in members in high-emitting sectors, focusing on them might achieve the right balance between tractability and materiality.”

Similarly, the **introduction** addresses the topic better:

“Similarly, qualitative assessments of transnational climate initiatives have highlighted how other issues may exacerbate data problems, such as lack of institutional capacity due to inadequate staffing or funding [3], or lack of quantifiable targets [12]. The degree of additionality of transnational initiatives remains a contentions topic due to the previously mentioned problems, raising concerns of double-counting, effort fragmentation and emission leakage effects [4, 6].”

Similarly, it would be helpful to see more critical engagement with the possibility that declining energy intensity/changing composition of business activities/declining carbon intensity of electricity have allowed companies to claim responsibility for the observed reductions where in fact they would likely have occurred anyway due to wider trends in, for example, the cost of renewable energy, the rules governing dispatch of electricity in power markets, etc. The lack of clarity over this issue means claims that the observed trends in the data "demonstrate that the target setting approaches set by each initiative are sound" (p.8) are unconvincing.

These types of statements have been fixed.

We now say that "members are following the target-setting approaches required by each initiative, even if they vary in complexity", and go in-depth in the **discussion** about how these might still lack impact due to intermediate steps.

"However, achievements may not directly translate into additional emission reductions. Several recent studies evaluating corporate action have shared concerns about the additionality of action due to low transparency [19, 30] and the lack of a robust mandatory reporting framework [18]. These issues were forewarned by studies expressing worries over the additionality and quality of initiatives outside the UNFCCC [5, 6]. Also, the UNEP assessment of the impact of international cooperative initiatives for 2020 [48] already assumes that about one third of the emission reductions achieved would not be additional to national climate action. Given the fact that a substantial part of the actions implemented were through renewable energy sourcing, and this is for more than half done by using low-additionality sourcing methods, it can be expected that until 2019 the overlap with national climate action is higher than what was suggested in the UNEP report.

Separating an initiative's progress from national trends is not straight forward process since lack of action by other actors may counteract their benefits [4]. Determining the level of additionality more accurately would require in-dept investigation at the company level, or comparison participating and unengaged businesses in different sectors and regions, which are clearly important next research steps. Since our results reveal that mitigation progress is heavily concentrated in members in high-emitting sectors, focusing on them might achieve the right balance between tractability and materiality. For initiatives such as the SBTi and RE100, a crucial step to increase additionality would be to move to more high-quality renewable energy sourcing methods, such as PPAs and self-generation. And anyway, more transparent disclosure and a better evaluation of accounting procedures are prerequisites for a credible contribution by companies to global emission reduction [30]. Without this step, evaluation of other metrics such as Scope 3 mitigation will remain difficult. Further research into this Scope, both in terms of fitness of target coverage and the robustness methods used to calculate and verify these emissions, might prove to be crucial in understanding whether companies in initiatives affect value-chain behaviour."

- Excluding Scope 3 emissions is a major omission since it comprises the bulk of many large companies' emissions. While this omission is understandable due to data limitations, the authors should be more upfront about this in the abstract and introduction, noting that their results are confined to direct emissions and emissions from energy purchases and exploring the implications of only looking at Scope 1 and 2 in more detail, rather than referring to this in the last sentence of the paper without any analysis of how this limits or qualifies the findings in the paper.

We agree that it is important to discuss why Scope 3 was omitted. We now include a thorough explanation in the introduction.

Introduction:

"The study only concerns emissions accounted as Scope 1 and Scope 2, which are direct emissions produced within company boundaries and indirect emissions from energy purchases, respectively [28]. We do not evaluate targets related to Scope 3, which accounts for emissions from upstream

and downstream activities [29]. Although emissions in this scope are often larger than Scopes 1 and 2, by definition Scope 3 emissions come from third parties in the value chain of a company, meaning targets would require influencing the behaviour of other businesses [30] that might not be members in either initiative, while Scope 1+2 are always within a company's sphere of influence. Second, all primary targets in both SBTi and RE100 concern Scope 1+2, while only SBTi members whose Scope 3 emissions account for more than 40% of the total are required to set targets for this scope [18]. Third, there is an absence of consistency in the methods and data used to account for Scope 3 emissions [31], making cross-company comparisons difficult. Finally, evaluating if initiative targets lead to companies implementing changes gives crucial information on the legitimacy of their commitment. If Scope 1+2 action is ineffective despite being within the operational reach of these companies it is unlikely that Scope 3 actions will fare better.”

- The claim in the discussion section "that the target setting approaches set by each initiative are sound" does not necessarily follow from the results (see general comments). Why does the fact of emissions reductions ahead of these targets demonstrate that the approach is sound? Please consider re-wording less strongly and acknowledging more clearly that there are reasons why the targets have been met that may be unrelated to the initiatives and the target setting approach itself. This is a subtle but very important nuance as this sort of phrasing can give the misleading impression that the initiatives have not only successfully corralled companies into setting targets but that this process was the only relevant factor in achieving them, which the authors have not shown convincingly. We agree that this framing was not ideal. It has been fixed in the new version with better phrasing to avoid giving the wrong impression on the study's outcomes.

We now say that “members are following the target-setting approaches required by each initiative, even if they vary in complexity”, and go in-depth in the **discussion** about how these might still lack impact due to intermediate steps.

Similarly, the discussion section is more thorough in the topic of additionality and its challenges.

And as the discussion goes on to say on p.9, the demonstrated reductions have taken place among a handful of companies, with little evidence that the target-setting exercise has achieved anything much beyond increased renewable energy procurement, which in any case may well have happened irrespective of target-setting (see general comments on additionality). This finding, i.e. that the vast majority of emissions reductions in this analysis are due to renewable energy generation and procurement either directly, or more often through instruments of varying quality, and very little fundamental (Scope 1) or cross-value-chain (Scope 3) progress has been made on emissions, could be made more prominent as a central result of this analysis that would also make clearer the limitations of the data and climate actions taken.

The abstract and discussion have been improved, specifically highlighting how most actions are only contractual in energy purchasing, with a clear preference to low-quality instruments. Although we do not feature Scope 3 results as they are not a key goal of the study and due to our wish to highlight other problems in the initiatives, we have integrated known issues of this aspect of corporate action in the introduction and discussion.

Abstract: “However, most of these reductions are concentrated in a small number of intensive companies. Most members show little evidence of emission reductions within their operations, only achieving progress via renewable electricity purchases. We highlight how intermediate steps regarding data robustness and implementation of sustainability measures are lacking: 75% of public company data has only limited independent verification, and 71% of renewable electricity is sourced through low-impact or unknown sourcing mechanisms.”

Results:

Substantive progress: “All sectors except Transport (n = 2) show this trend, with Electricity Generation companies (n = 3) being responsible for 78% of the over-achievement. However, collective 2015 emissions exceeded the sum of all absolute target baselines by 36 M tCO₂e. This suggests that overall emission reductions achieved in this group may be a continuation of preestablished trends rather than an acceleration of mitigation action. A similar behaviour can be seen in SBTi members with intensity targets (n = 7), which showcased steady emission decreases at a faster AAGR of -12.6%.”

Discussion: “Separating an initiative’s progress from national trends is not straight forward process since lack of action by other actors may counteract their benefits [4]. Determining the level of additionality more accurately would require in-dept investigation at the company level, or comparison participating and unengaged businesses in different sectors and regions, which are clearly important next research steps. Since our results reveal that mitigation progress is heavily concentrated in members in high-emitting sectors, focusing on them might achieve the right balance between tractability and materiality. For initiatives such as the SBTi and RE100, a crucial step to increase additionality would be to move to more high-quality renewable energy sourcing methods, such as PPAs and self-generation. And anyway, more transparent disclosure and a better evaluation of accounting procedures are prerequisites for a credible contribution by companies to global emission reduction [30]. Without this step, evaluation of other metrics such as Scope 3 mitigation will remain difficult. Further research into this Scope, both in terms of fitness of target coverage and the robustness methods used to calculate and verify these emissions, might prove to be crucial in understanding whether companies in initiatives affect value-chain behaviour.”

Specific comments

p.1

- "Qualitative assessments have shown other problems that exacerbate data issues, such as lack of permanent staff, adequate funding or measurable goals". Does this refer to staff/funding/goals of voluntary platforms, or to resources committed voluntary initiatives within companies? May be worth clarifyin and/or expanding.

It speaks of the second (initiative resources). The argument has been separated for each referred paper: Chan speaks of institutional capacity while Michaelowa talks about targets being measurable.

“Similarly, qualitative assessments of transnational climate initiatives have highlighted how other issues may exacerbate data problems, such as lack of institutional capacity due to inadequate staffing or funding [3], or lack of quantifiable targets [12].”

p.2

- Regarding the NCI and SBTi studies, it is also worth noting that there is not necessarily any evaluation in those studies (unless the authors know otherwise) of whether the targets are appropriately ambitious, which adds to the problems of insufficient transparency and the widespread use of poor-quality offsets and renewables instruments.

This has been clarified in the paragraph to provide a more complete picture. The lack of target trend comparisons has been highlighted in the paragraph.

The few available analyses give mixed results, with a recent report by the NewClimate Institute et al. [17] stating that 80% of the targets set by 119 companies are on track to be overachieved, and Gieseckam et al. [18] showing that 74% of primary targets in 81 companies in the Science-Based Targets initiative (SBTi) are either on-track or surpassed. However, these percentages include large

and small companies and cannot say much about quantitative impact, and target ambition is not compared against necessary global trends to achieve Paris goals.”

- The statement on the Lui et al study is missing key pieces of information required to assess its validity. Suggest remarking on how much of an issue ignoring overlaps (i.e. double counting) is, and how large each of the hypothetical 2000 members would have to be in terms of emissions footprint for their estimates to be met.

An additional statement was included to make the limitations of those amounts explicit.

“Lui et al. [9] estimated that both initiatives can significantly reduce global emissions beyond NDCs if they grow to 2000 members by 2030: 2.7 GtCO₂e for SBTi and between 1.9–4.0 for RE100. However, these amounts may not manifest due to overlaps in membership, which already occur [20], and if the average emissions of new members are lower than the current mean.”

- Please provide more information, either here or elsewhere in the paper, on how valid the qualification process is, and how robust SBTi's method for assessing consistency with 1.5C/2C pathways is.

SBTi's method was made more explicit to showcase the differences in the approval process of absolute targets and intensity targets. However, further explanations of SBTi's methodology are beyond the scope of this study. We have added references to other papers explaining SBTi's methods to account for this.

“The SBTi was created in 2015 by a coalition of CDP, the United Nations Global Compact, the World Wide Fund for Nature and the World Resources Institute, focusing on evaluating GHG reduction targets set by companies. Committed members are subjected to an approval process [23] comparing their target’s trend against integrated assessment model scenarios that keep global emissions at least below 2°C or a more ambitious 1.5°C (Absolute Contraction Approach [24]), or against sector-specific intensity trends keeping emissions below 2°C (Sectoral Decarbonisation Approach [25]).”

- It is not clear what the "two year waiting period" in the SBTi contact refers to. Please clarify.

Improved the statement by explaining further.

“However, the distribution of qualifications among members with approved targets was similar: more than half reached 1.5°C (53%), 25% met well below 2°C, and the rest remained in the obsolete 2°C qualification. In the case of members committed to setting targets, SBTi specifies a two-year limit to get targets approved and published before a company is removed from their listings [32]. Despite this rule, 24 of the 44 committed members were already past the maximum, with 14 being in the financial sector. RE100 membership was lower at 68, all with targets set, representing 24% of the initiative’s membership (289, as of February 2021).”

p.3

- Authors refer to "financial businesses with low emissions" in RE100. Does this include financed emissions in their portfolios?

It does not include portfolio emissions, since RE100 only targets electricity consumption. To make this clearer, we have changed the statement from "low emissions" to "comparatively low energy use".

“By contrast, RE100 had a significant amount of exclusive members (i.e., those participating in a single initiative) in the Services sector (n = 17), primarily financial businesses with comparatively low energy use (n = 15).”

- Please explain why fossil fuel producers can commit to SBTi targets but they cannot be approved. Is it because SBTi does not engage with these companies or is it a question of measurement?

At the time of the study, SBTi only allowed such companies to commit “symbolically”, and did not formally approve any of their targets. We have made the statement clearer for both initiatives.

“No Fossil Fuel Production companies participate in either initiative. At the time of this study, SBTi only allowed these companies to commit to the initiative but did not formally approve any of their targets, and RE100 explicitly disallowed energy producers from becoming members [26].”

p.4

- First sentence is long and grammatically confusing - please revise. Perhaps "falling outside...2015-2019" should be placed in parentheses and semicolons used to separate list items. Please also explain more clearly why 2019 is the cutoff point.

The statement was improved to reduce repetition, clarify better what Scope 1 and Scope 2 are, and to justify the 2019 cut-off.

“Our analysis only covers emissions and energy use within company boundaries (Scope 1) and those related to energy purchases (Scope 2) [29, 33]. Targets covering the supply chain and end-use emissions (Scope 3) of members had to be excluded, as set out in the Introduction section. The evaluated period was from 2015 to 2019, the last year for which complete CDP and sustainability report data was available. It includes only RE100 participants who joined at or before 2019 (n = 58) and SBTi members whose target baseline was set at or before 2019 (n = 70).”

- What does "evaluated partially" mean with reference to intensity targets, and why do they suffer more from "disclosure problems" than absolute targets?

The statement was clarified. The only metric missing for these companies was the estimated target trend. Intensity targets require activity metrics (which can be based on company output or financial performance). Giesekam et al. highlighted that this can cause consistency issues because data sources external to the company are required in some cases.

Our analysis only uses data directly disclosed by members (either CDP or yearly reports) so we opted to skip this step for these members.

“For simplicity, target trends for SBTi companies with intensity targets (n = 7) were not projected. These targets require different metrics of company activity (e.g., tons of cement or kWh generated per year), which suffer from additional disclosure problems and often require consulting data sources unrelated to the company [18].”

- Why are emissions held constant for years before the baseline, if not a lack of data? This might give the misleading impression of constant historical emissions for a company for whom emissions were already falling for other reasons (as is pointed out on p. 7).

Emissions were held constant because the Ambition metrics only relates to data about the targets themselves, and not historical emissions. Doing otherwise could give the impression that targets were in place before the baseline year, when it is not the case.

Regarding the concern of misleading impressions, it is why the same target trends were included in figure 6a (where the baseline emissions were also held constant for the targets).

- Please explain in more detail, or refer to further methodological documents, what you mean by comparing company-level trends to "normalized scenarios for OECD+EU nations". Does this mean that the additionality of emissions reductions is being calculated against economy-wide emissions trends? This is probably not a fair comparison as the companies represented in the same may tend towards less emissions-intensive activities than the wider economy although it appears that your assessment of consistency with SSP1 is based on sector-specific pathways. If this interpretation is not correct, please clarify.

As said, we compare against economy-wide emission trends (now for both OECD and World). However, we are careful not to state that this says anything about the additionality of the targets themselves, only the ambition of the trend they aim for. This was necessary since there is not enough data to know the geographical distribution of the company's operations, the exact type of activities carried within those facilities, and the ambition of the government (and other actors) in that space.

Projecting their trends against regional or global scenarios aids in understanding the target's relevance, and in evaluating the target-setting process of the initiatives.

Regarding the Scenarios, we only compare against regional/global scenarios. The IIASA database did not include sectoral pathways for these models at the time of the study. These particular scenarios were selected due to being the most ambitious pathway in the SSP exercise, and because the SBTi itself already uses them as part of their Absolute Contraction Approach.

“Targets were then compared against scenarios taken from the IIASA database for the IPCC Special Report on Global Warming of 1.5°C [34]. These were selected based on relevancy by ensuring they did not assume a global decrease in emissions between 2010–2020, which did not occur [55], that they were used in the SBTi’s Absolute Contraction Approach [23, 24], and that they were consistent with highly ambitious shared socioeconomic pathway narratives for achieving Paris goals [35]. The selected scenarios were SSP1-19 (consistent with 1.5°C) and SSP1-26 (consistent with below 2°C) produced by the AIM/CGE 2.0, GCAM 4.2 and WITCH-GLOBIOM 3.1 models.”

Please also extend this clarification to explaining how the baseline figure is calculated - is this done consistently across all companies on the basis of a single baseline year?

The target baselines were taken directly from these documents:

- SBTi's "companies taking action" database
- RE100's 2020 progress report.

However, SBTi does not state the baseline in absolute amounts. To correct this lack of data, we analysed the most recent CDP responses of SBTi members, since they include an option to state SBTi targets. Target baseline emissions were taken from this section.

If a company submitted a baseline that exceeded actual historical emissions, or if it did not disclose the baseline at all, it was assumed that it covered 100% of the actual historical emissions of the baseline year.

This was done consistently for all members.

“Target data had to be obtained differently for each initiative. For the SBTi, it was obtained from the initiative’s website [58], and for RE100, the most recent progress report was used [20] (in the case of recent members, the RE100 website was used instead, although it tended to omit interim targets). At their most basic, targets consist of a baseline year, a baseline value (e.g., tCO₂e, renewable electricity ratio), a targeted reduction or increase, and a target year [24]. However, no initiative disclosed target data with complete transparency: baseline values were missing in the initiatives’

websites at the time of the study, or they were given in percentages in annual reports [20, 21], which made the impacts of the commitments indistinguishable between members. In the case of absolute targets in the SBTi (i.e., targets tracking only tCO₂e emitted) and all RE100 targets, CDP questionnaires were used to establish baseline target values in more concrete terms (either tCO₂e emitted in that year or total kW h of renewable electricity consumed). Data related to Intensity targets in the SBTi, which follow the Sectoral Decarbonisation Approach [25], was not collected because recent studies have highlighted that these targets suffer from additional transparency issues and often require data from unofficial sources [18] and this study focuses only on data officially disclosed by companies in our sample.”

p.5

- Figure 3 - the right hand panel (b) doesn't seem very necessary or additional to (a). Suggest keeping just (a).

We thought it would be useful to also provide absolute numbers. The box plot in the left figure tells us something about the distribution of the targets, but not about the weight of the individual companies. Moreover, most initiatives themselves just present relative numbers, so this is also an important addition to the public information available.

p.6

- Why did CDP bar financial companies from disclosing RE purchasing methods?

We have added a citation to CDP's questionnaire changes documentation for 2020, to show that the change was not accidental. The reasoning behind this change is unknown to us.

“This decrease is primarily driven by recent changes in CDP, which barred Financial companies from disclosing their purchasing methods in the questionnaire [43], explaining why the decrease is mostly seen in RE100 since it has a high amount of members in the Financial sector (n = 22).”

p.7

- Please explain more clearly why U-EACs are considered lower quality than PPAs and why their displacement of FF sources and emissions reductions attributes are different to those of PPAs. The logic is not made explicit. If the logic is indeed sound, the authors should draw attention to most of the RE procurement being of this lower quality, but also consider why some companies might choose U-EACs over PPAs: are they cheaper, for example? If so, why?
This oversight is now provided in the methodology section.

“Second, by disaggregating purchased renewable energy into four common sourcing methods with different degrees of potential additionality [45]. Unbundled Energy Attribute Certificates (U-EACs, also known as Renewable Energy Certificates) are decoupled from energy contracts [33], with producers deciding if they'd rather retire the credit themselves (thus lowering their emissions), or sell it to another party who can claim it as their own. They are considered to be of low additionality due to difficulties demonstrating their impact on new renewable installations [61], low prices [45] and short contractual periods [46]. Utility green products (Utility GPs) are sold by energy producers at a premium, and may come from installations built through public support schemes [45]. Power Purchase Agreements (PPAs) are long term arrangements between users and power producers which are generally seen as having a high potential to result in new renewable capacity installations [19, 45]. Finally, self-generated renewable electricity is seen as the most additional since it implies that the company invested in the capacity themselves [19].”

- In measuring the "rate of consumption of high additionality instruments per member" you should really be normalising against revenue or total electricity consumption, since without this, variance

in total RE purchased between RE100 and SBTi could just be explained by differences in company size/emissions intensity (and perhaps it isn't but we can't tell from the data provided)
To show the differences in high quality sourcing, we now show the ratio of high quality instruments against total energy purchases. The quantities shown have been updated in the paragraph.

“However, the total sourced renewable energy is similar in the three groups (Figure 5b), with RE100 being a clear determinant of better renewable sourcing practices. Comparing high-quality sourcing against total purchased energy in 2019, the RE100 exclusive group had the highest share of high additionality instruments at 22.1%, while its SBTi counterpart had the lowest at 4.1% and overlapping members had a slightly better 8.5%.”

- Please reconsider the word choices "good trends" and "impressive" in relation to AAGRs for SBTi members with intensity targets. These are subjective assessments and should be treated more objectively in this paper.

We thank the reviewer for this insight. All statements have been changed to show a more neutral and consistent tone.

- Please clarify whether the disparity between RE100 reports and public disclosure indicates an underreporting or overreporting of RE consumption. Currently the authors use the word "offset", which is ambiguous.

We have made the statement clearer: it was an over-reporting by some members.

“In the case of RE100 (n = 58), several companies over-reported their renewable electricity usage to RE100 in the early years of the initiative, affecting target baselines in 2015 and producing an offset between targeted renewable consumption and actual renewable sourcing of 19 T W h by 2019.”

p.8

- It is not clear from the results whether or not Scope 1 electricity sector emissions reductions are being double counted in the form of Scope 2 emissions reductions in energy intensive industry or other sectors that consume this electricity.

We are unable to separate Electric Utilities from the rest of the companies. To do so would require full knowledge on the energy contracts of end-use companies. To address this comment we have made this limitation explicit.

Results:

“Substantive progress indicators show decreasing GHG emission trends in both initiatives (Figure 6). Assuming no overlaps between utility companies and other members, GHG emissions of SBTi participants with absolute targets (n = 63) have an AAGR of -7.8% and surpassed their targetted reductions by 34 M tCO₂e in 2019.”

Limitations:

“Fifth, it is possible that overlaps exist between Electricity Generation and other sectors, reducing mitigation and overall initiative effectiveness. Public information is insufficient to identify such cases. A more targeted study, perhaps evaluating the contractual preferences of high-emitting participants, would aid in addressing this gap.”

p.9

- Spelling error: "where utility companies" should be "were utility companies"
Corrected.

- Suggest qualifying the claim that "companies have successfully met the goals established by these two initiatives". This study does not show this; it shows that a subset of disclosing firms (who, by disclosing their emissions, is probably biased towards those more likely to be achieving their

targets) are on trend to meet targets that are for 2030 (RE100) and targets that are set and verified by the companies themselves, not directly (SBTi).

The language on the discussion section has been revised to be more neutral.

“These trends show that members are following the target-setting approaches required by each initiative, even if they vary in complexity”

Reviewer #2 (Remarks to the Author):

I enjoyed the material presented in the paper and believe, on the basis, of the analysis that the materials accurately reflect what is transpiring in the initiatives you have tracked. There are, however, matters of framing that are likely to fundamentally reshape how you might understand the data you present. Both of these comments require a more nuanced view of corporate functioning, carbon accounting and accountability. It is perhaps an omission to have conducted this work without the obvious input of a specialist in this area. The two issues of framing then ...

We thank the reviewer for their comments.

We believe that there is enough experience with scenario analysis, transnational initiative studies and company target setting within the team of researchers. The second author has published a variety of papers on these topics, see for example: <https://doi.org/10.1038/nclimate2770> and <https://doi.org/10.1016/j.jclepro.2014.09.046>. He has also worked as a consultant and advised many large companies and governments on these issues.

1. You state in the opening of the paper that contributions by corporations to emissions reductions will be additional to NDCs. This is not likely to be true because the choices that corporations make for reductions are set within the context of country actions to employ policy levers to reduce emissions. Many of these emission reductions, then, will be counted by both countries and corporations. At the end of the day, it is the country reduction trajectories that matter with the picture of corporate reductions providing a different 'cut' through much of the same activities. To spell this out a little further. If a country has an emissions reduction programme in place that requires companies (as the subject of regulation) to reduce their emissions then your rationale would count these reductions twice – when in fact they are one in the same reductions, just with different entities articulating their timing and nature.

We acknowledge the risk of double counting and that targeted action at a government level will have to be adopted at the company level. However, the truth is that NDCs are not ambitious enough. The primary reasoning behind the analysis of non-state action is the "emissions gap" between NDC submitted by governments and Paris compatibility (<https://doi.org/10.1038/nclimate1602>).

Even if a government has set a target nationwide, that does not impede regions or companies from adopting more stringent measures. That is when analysis of additionality comes in (see Fig 1 in this paper by Hsu on how additionality of city targets might be measured <https://www.nature.com/articles/s41558-018-0338-z>). IRENA also acknowledges this in some of their literature on energy sourcing (<https://irena.org/publications/2018/May/Corporate-Sourcing-of-Renewable-Energy>).

A good example is the US after leaving the Paris agreement. Several subnational and private actors decided to maintain or even increase their targets even after the country disengaged from the agreement (<https://newclimate.org/sites/default/files/2017/09/states-cities-and-regions-leading-the-way.pdf>).

2. At the top of page 5 you note that the reductions 'exceeds the ratio for OECD + EU nationals where most of these companies have their headquarters'. This seemed a very odd comparison to make. The companies you have in the same are large and therefore likely to have operations in many different countries. Their emissions totals, therefore, do not necessarily have any connection to their headquartered countries because they will be the sum of emissions in a number of countries. This makes comparing reduction ratios non-sensical.

There are two extremes regarding this: all company operations are within OECD countries or distributed equally around the globe. We acknowledge that it is likely that reality sits somewhere in the middle. However, the nations where these companies are headquartered might have the more

institutional capacity to enact changes in their behaviour if they so choose, so the OECD simplification seemed the most adequate. We have added similar graphs (at a global scale) to the supplementary information for completeness.

Results:

“Although most of these companies are headquartered in OECD countries, the distribution of their operations might differ. To account for this, we compared targets against global scenario trends (see Figure 3 and Figure 4 in the supplementary material). Doing so extends the allowable range for 2°C compatibility in the models selected for this study (see Methods) and reduces the acceptable upper range for 1.5°C. Most RE100 members still outperform 1.5°C conditions due to the initiative’s requirement of 100% renewable electricity by 2050. However, SBTi is no longer collectively within 1.5°C due to an excess of 18.2 M tCO₂e. Services was the only sector that maintained 1.5°C trends, with all others only reaching below 2°C ambition.”

Supplementary:

“For completeness, here we present ambition against the global scenarios. Figure 2 and Figure 3 in the study describe a case where companies operate primarily within the OECD. However, large international companies also operate outside of this group of nations. We opted to present OECD trends since the headquarters of these companies are within that group, meaning they might have more institutional capacity over them, and because a fair amount of their operations are likely still concentrated within those nations.”

Supplementary Figure 3: **Ambition among SBTi members with absolute targets against global SSP scenarios.** *t* represents the number of companies with active targets in a year. Targeted emissions were kept constant for years before and after a company’s target baseline and deadline, respectively.

Supplementary Figure 3: **Ambition among SBTi members with absolute targets against global SSP scenarios.** t represents the number of companies with active targets in a year. Targeted emissions were kept constant for years before and after a company's target baseline and deadline, respectively.

Supplementary Figure 4: **Ambition of RE100 members ($n = 58$).** **a)** Box plot distribution of the renewable electricity ratio targeted by each company compared against global SSP scenarios of the ratio of renewable electricity. **b)** Growth of total renewable electricity. t represents the number of companies with active targets in each year. Companies that had not joined the initiative are excluded in the early years.

Alongside these framing issues there are other aspects that could usefully be addressed:

- While I can appreciate why scope three was outside of the analysis (at least because of the inconsistent way in which such measures are made), scope 3 emissions for any individual company are frequently many factors greater than scope 1 and 2 emissions. Some way of acknowledging this fact would be helpful and what it means for the thrust of your argument are non-trivial. We now address our reasons for omitting this scope in the introduction. The key issue with them is their non-standard calculation methods, necessity to influence the behaviour of companies outside the operational control of members, and data problems.

“The study only concerns emissions accounted as Scope 1 and Scope 2, which are direct emissions produced within company boundaries and indirect emissions from energy purchases, respectively [28]. We do not evaluate targets related to Scope 3, which accounts for emissions from upstream and downstream activities [29]. Although emissions in this scope are often larger than Scopes 1 and 2, by definition Scope 3 emissions come from third parties in the value chain of a company, meaning targets would require influencing the behaviour of other businesses [30] that might not be members in either initiative, while Scope 1+2 are always within a company's sphere of influence. Second, all primary targets in both SBTi and RE100 concern Scope 1+2, while only SBTi members whose Scope 3 emissions account for more than 40% of the total are required to set targets for this scope [18]. Third, there is an absence of consistency in the methods and data used to account for Scope 3 emissions [31], making cross-company comparisons difficult. Finally, evaluating if initiative targets lead to companies implementing changes gives crucial information on the legitimacy of their commitment. If Scope 1+2 action is ineffective despite being within the operational reach of these companies it is unlikely that Scope 3 actions will fare better.”

- At page 3 you note that Chinese companies stood out as absent – which is not surprising. While their companies may be very large, they often have a significant percentage of ownership with the Chinese state. Where this is the case, their propensity to participate in voluntary corporate initiatives will be limited. This is a common observation in the accounting/business/management literature – that Chinese corporations work within very different sets of norms and (more generally) that corporate ownership profiles generate different propensities for pro-environmental behaviours and reporting practices.

Previous studies have found that engagement in China in transnational climate initiatives is not insignificant. We have added a citation to the paper that prompted our comment, and corrected it to make it more neutral sounding.

“Low levels of engagement ($\leq 20\%$) were primarily seen in Asia. Chinese companies were mostly absent even though previous studies on transnational climate participation have found some degree of engagement within the country [1]. China is the largest nationality in the G500 ranking (124 total), but only one was present in any of the initiatives.”

- There is a need to better understand why limited assurance is likely for voluntarily disclosed carbon information. The issue here is one of costs and benefits as well as the complexity of the greater level of data certainty required for what is often unregulated measurement protocols (including estimations from emission factors cf direct measurements). Moreover, the highly distributed data sets that will usually sit behind Fortune 500 company reports make limited assurance the most suitable form. It is notable in your data that those sectors (eg utility companies in countries with high levels of regulatory oversight) are able to subject to reasonable assurance. This is a function of underlying data complexity and capture.

Lack of regulation nor “data complexity” should not be excuses here. Systematic complexity, specially in relation issues that affect health, the environment and climate, should be met with more careful reviews and processes, not less.

Stakeholders also acknowledge the need for better practices, although companies oppose it.

see: [https://eur-lex.europa.eu/legal-content/EN/TXT/PDF/?uri=PL_COM:Ares\(2020\)3997889&from=DE](https://eur-lex.europa.eu/legal-content/EN/TXT/PDF/?uri=PL_COM:Ares(2020)3997889&from=DE) (cited in the paper)

- You note that there are year by year inconsistencies in CDP data but I am not sure what you mean by this. The standards and approaches are not likely to be consistent as measurement protocols (specified by countries) can change, emission factors change as energy systems are transformed, figures may vary depending on changes in corporate activities and the CDP itself changes its information needs. This co-evolution of these factors will mean that consistency is probably not available as the approaches reflect underlying changes.

We acknowledge that environmental data is heavily affected by underlying changes a company's structure and performance.

However, most of the data issues encountered by the study have nothing to do with this, and instead were caused by incorrect data capture, calculation mistakes or by misunderstanding GHG Protocol guidelines.

For example, using the incorrect order of magnitude, ignoring changes in CDP questionnaires, etc. Such obvious mistakes are unlikely to be caused by a firm's evolution.

Supplementary material: added as section detailing several of the problems seen in CDP responses and company reports.

- You note that membership of the initiatives is concentrated on OECD countries – which is a reflection on your data being based on the Global 500 list. This observation is driven by your

sample design and hence its relevance to a line of argument is uncertain.

It is China that is overrepresented in the G500, not OECD nations.

The sample design was driven by the fact that revenue (which the G500 is based on) is a better indicator of firm size than Market cap. This choice was driven by materiality (high revenue members likely cover more emissions than low revenue ones).

Also, reports by the initiatives themselves and scientific literature have acknowledged that participation is concentrated in OECD. This comment is meant to highlight that our study is representative of the initiatives' composition.

Results: added citation to SBTi and RE100 reports.

“Finally, it is noteworthy that membership is concentrated in OECD nations, which mirrors the results of previous studies looking into the distribution of transnational climate initiatives [1, 5] and initiative reports [20, 21].”

- In places you talk about improved, centralized and converging disclosure practices ... what might be of more relevance is measurement protocols on which disclosure is based. This confusion is also obvious at the topic of page 2 when you say that the GRI leaves plenty of leeway for how companies account (ie measure) and publish (ie reporting) environmental data. It is worth noting that, most usually, the WRIs GHG Protocol is the measurement tool (unless country regulations stipulate a different measurement) while the GRI is a reporting standard.

This is correct. However, most companies are already using the GHG Protocol for measuring their emissions (CDP questionnaires are based on it). However, using the same measuring method does not guarantee that the information will be compatible between, say, a GRI report and CDP. In many cases companies decided to omit a GHG scope from the reports entirely, presented the data in non-comparable ways (percentage vs absolute), varied the reporting boundaries between reports or years, etc.

To address this, we have improved the paragraph mentioned by the reviewer to make the meaning of convergence and comparability more explicit. The mention of "accounting" was removed for GRI since, as the reviewer rightly points out, it is a reporting standard and not one that deals with measuring itself.

“In the case of companies, evaluation is complicated further by a lack of convergence in how data is presented, driven by a plurality of reporting standards, disclosure platforms, legislative differences and conflicting stakeholder priorities [13–15], with even widely used reporting standards such as the Global Reporting Initiative (GRI) leaving some leeway on how environmental information is disclosed. Similarly, submissions to more centralised sustainability reporting platforms such as CDP (previously the Carbon Disclosure Project) may be incomplete or affected by year-by-year inconsistencies, which complicate longitudinal evaluations [11, 16]. This lack of convergence means that, even for the same company and year, the information in a CDP response and a sustainability report published by the company might not be comparable due to data omissions, differing calculation boundaries or methods, or incompatible presentation (e.g., disclosing emissions in absolute numbers or as mere percentage decreases).”

- On page 9 in the second to bottom paragraph you refer to studies [11, 15, 18, 48]. I can't locate reference 48 in the list and 11 and 15 are not obviously references for the point you are making. Both Kolk[11] and Matisoff [15 (now 16)] talk about how frequent changes in the questionnaire complicate longitudinal analysis. In our mind this is very related to the issue at hand.

Discussion: the argument was separated in two parts to explain the references better.

“Similarly, there are still plenty of problems obstructing the appropriate use of CDP data, such as changes in CDP guidelines complicating longitudinal comparisons [11, 16] and active omissions by the companies themselves [19, 51].”

Link to Stanny (ref 48 in the comment):

<https://www.tandfonline.com/doi/full/10.1080/0969160X.2018.1456949>

DOI: <https://doi.org/10.1080/0969160X.2018.1456949>

Reviewer #3 (Remarks to the Author):

Key results

The manuscript investigates ex-post the effects of two important climate initiatives, SBTi and RE100, and the progress of their participating organisations on Scope 1 and Scope 2 climate actions, from 2015 to 2019. Notably, the study highlights that collectively all analysed SBTi organisations are aligned with IPCC pathways for well below 2°C, whereas REC100 organisations show a share of renewable electricity consistent with well beyond 1.5 °C requirements.

Surprisingly, organisations have indeed reduced their Scope 1 and Scope 2 CO₂ eq. emissions by 35% compared to the starting baseline. However, the bulk of this result was achieved by a small group of five utilities and energy-intensive companies, accounting for 85% of the total reduction. Therefore, the study shows mixed results once a more in depth-analysis is carried out and this is extremely interesting as it sheds lights on some numbers that otherwise would be misinterpreted. The same goes for renewable energy adoption.

Although the share of renewable purchasing is increasing, the quality of purchasing instruments do vary. In addition, the study provides further insights compared with previous literature by assessing organisations on their ambition, methodology, implementation, and progress on climate actions. Overall, the study offers updated and useful data and information to evaluate and monitor the validity of those two relevant on-going climate initiatives on scope 1 and scope 2 emissions, also assessing organisations progresses in reaching science-based targets.

We thank the reviewer for their comments. We agree on the usefulness of our data and methods for evaluating corporate commitments to tackle climate change.

Validity

Data interpretation and conclusions are sound. Strong evidence is provided for the authors' claims and all appropriate controls have been included. However, the paper missed a data validation test on energy intensive and utilities organisations.

As a small clarification, only utilities suffer from additional disclosure issues. However, due to the importance of the five utilities to overall initiative progress, we have re-analysed them and updated our data, including an evaluation file in the public databases included for this study, and a clarification in the methodology detailing issues seen and our approach to evaluating these companies

After this revision, the overall contribution of these eight high emitter has increased to 86%. We have updated the data in the study to reflect this small change.

Methodology:

“Only energy end-use companies could be tested thoroughly. Our tests could not apply to companies in the Electricity Generation sector due to unique disclosure issues. First, CDP questionnaires for 2015–2016 did not provide crucial data such as gross/net generation, installed capacity or per-technology emissions. Second, Scope 2 emission calculations have additional complexities due to transmission losses being included within them if the company also has a transmission or distribution business [33]. Lastly, self-released reports by these companies did not provide enough detail to construct the energy consumption section. Instead, the emissions of these companies were compared against the amounts stated in sustainability reports to identify mistakes during CDP submission. A detailed document for this sector is available in the database, explicitly stating the reviewed documents as well as issues that were identified and corrected.”

Significance

The present work is extremely interesting for both the literature and practitioners.

On the one hand, the few ex-post studies on the literature focus on the achievement of SBTi targets by mixed small and large companies, but they do not focus and express the total impact (quantitative data) of large organisations. Differently from previous literature that expresses a certain percentage of analysed organisations to be on track with SBT and IPCC requirements, this study evaluates altogether the efficacy of both initiatives to match such requirements by studying the total amount of CO₂ emission reduction and the increase of renewable energy usage of the analysed large organisations. It provides also further insights than the previous literature due to the evaluation of the robustness of the verification processes.

On the other hand, ex-post data analysis is extremely useful for practitioners as they can be used as monitoring tools but also to strengthen the initiatives themselves by focusing on the improvement needs and limits for monitoring highlighted by the paper.

However, the exclusion of scope 3, considering the sample of the study, might limit the relevance of this paper.

We thank the reviewer for this comment.

We also agree on the importance of Scope 3 emissions for overall company action. However, the omission of this Scope was through data issues that render it incompatible with this type of evaluation. Essentially, there is no way to evaluate the data, making emission increases or changes in a company's Scope 3 accounting method indistinguishable.

We now highlight these problems in the introduction:

“We do not evaluate targets related to Scope 3, which accounts for emissions from upstream and downstream activities [29]. Although emissions in this scope are often larger than Scopes 1 and 2, by definition Scope 3 emissions come from third parties in the value chain of a company, meaning targets would require influencing the behaviour of other businesses [30] that might not be members in either initiative, while Scope 1+2 are always within a company's sphere of influence. Second, all primary targets in both SBTi and RE100 concern Scope 1+2, while only SBTi members whose Scope 3 emissions account for more than 40% of the total are required to set targets for this scope [18]. Third, there is an absence of consistency in the methods and data used to account for Scope 3 emissions [31], making cross-company comparisons difficult. Finally, evaluating if initiative targets lead to companies implementing changes gives crucial information on the legitimacy of their commitment. If Scope 1+2 action is ineffective despite being within the operational reach of these companies it is unlikely that Scope 3 actions will fare better.”

Similarly, we consider that we have a good coverage in terms of the emissions within direct control of initiative's members:

“Compared to 2019 data published by both initiatives, our 102 company sample likely includes 59% of the 1.2 GtCO₂e coverage mentioned in SBTi reports [21] and 59% of the 289 T W h of electricity consumption in RE100 [20]. Companies not featured are either too small to be in G500 rankings or businesses excluded due to problems in their disclosure methods.”

Data and methodology

Data quality, methodology and quality of presentation are sound for the aims of the article. Details are properly provided for reproducibility; however single company excels could be useful to check on database quality.

The analytical approach properly follows the phases of data gathering, environmental data

validation, benchmarks and progress. All the phases are thoughtfully designed for the objective of the paper and to increase data quality as much as possible considering the intrinsic data problems. Supplementary materials provide understanding of the methodology and some results. A weblink for a public database with all datasets is fully available.

5.1 Data gathering

Data gathering approach and the selection of databases are appropriate. Geographical region and energy sector values were associated with GICS categories for worldwide recognition. Members identification was properly carried out avoiding naming issues and the complete set was further assessed for omissions.

Target data were obtained from the most recent sources. The usage of SBTi data only under the absolute contraction approach instead of sectoral decarbonisation ones is justified. CDP questionnaires were appropriate and relevant sources to collect data in order establish baseline target values in more concrete terms.

The whole methodology followed to gather environmental data for all SBTi and RE100 participants in the G500 is logic and sound, and clearly visible in supplementary materials. Database construction is made with CDP questionnaires, annual reports and sustainability documents to be able to obtain all relevant data since data lacking is one of the most important problems in environmental disclosure. The approach is appropriate. However, from the data sets it is not clear which data are obtained from which documents.

Unfortunately, our methodology did not include adding the source of data at an atomic level. However, we have updated the public database to include all the code and individual company files, where the sources used are listed. We hope this aids in increasing reproducibility.

Moreover, authors say that whenever possible, “Scope 2 data collected were of the same type as the emission target of the company”. Since the use of different emission factors between market-based and location-based might be extremely relevant for the count of the total amount of GHG emissions, it could be relevant to have an understanding of the percentage of location and market based used for the calculations. In fact, SBTi companies can choose either one or another approach to set scope 2 targets. Consequently, it is not clear in the text and figure 2 which data (LB or MB) have been prominently used.

To solve this difficulty we have added more information on the supplementary material, breaking down MB and LB contributions for Figures 2 and 6.

Supplementary Table 6: **Disaggregation of substantive progress by GHG Scope, in $MtCO_2e$.** The data featured in Figure 6d is separated by sector and GHG Protocol scope, including Location Based (LB) and Market Based (MB) approaches for Scope 2[4]. Due to inconsistencies in emissions reporting, MB data was preferred if available. Otherwise, LB data was used.

Sector	Scope	2015	2016	2017	2018	2019
Electricity Generation	S1	371.64	330.9	302.39	250.47	196.74
	S2 LB	11.68	9.23	3.58	3.42	2.52
	S2 MB	7.72	8.41	11.11	9.89	9.99
Energy Intensive Industry	S1	206.59	194.98	173	165.77	150.67
	S2 LB	4.82	4.76	4.69	4.48	4.01
	S2 MB	15.99	10.67	9.52	8.39	7.53
Light Industry	S1	46.91	47.32	46.02	48.09	44.83
	S2 LB	15.44	13.88	11.04	7.06	7.42
	S2 MB	45.42	46.59	40.87	41.78	36.49
Services	S1	16.39	16.48	16.3	15.38	15.14
	S2 LB	26.68	7.7	6.07	5.84	5.38
	S2 MB	29.61	44.56	38.92	36.3	31.56
Transport	S1	4.17	4.26	4.29	4.2	4.15
	S2 LB	0.09	0.07	0	0	0
	S2 MB	5.51	5.14	5.12	4.28	4.03
Totals	S1	645.72	593.93	541.99	483.92	411.53
	S2 LB	58.71	35.65	25.38	20.8	19.33
	S2 MB	104.25	115.37	105.53	100.63	89.6

“Contrary to previous studies, our analysis has a special focus on the effects of collective action in the initiatives. Regardless, the individual performance of members against their targets remains an important aspect of non-state action studies. The following table discloses member progress against the extended linearised targets seen in Figure 2, Figure 3 and Figure 6.”

Supplementary Table 7: Disaggregation of target progress per initiative. In the case of the SBTi progress has been broken up by scope, including Scope 2 category.

Initiative	Companies	N. on track	% on track	Target unit	Total 2019	T. on track 2019	% on track
SBTi abs. S1	63	46	73%	$MtCO_2e$	157.58	134.75	86%
SBTi abs. S2 LB	16	10	63%	$MtCO_2e$	27.62	19.19	69%
SBTi abs. S2 MB	47	36	77%	$MtCO_2e$	56.12	33.68	60%
SBTi int. S1	7	-	-	$MtCO_2e$	246.14	-	-
SBTi int. S2 LB	1	-	-	$MtCO_2e$	2.52	-	-
SBTi int. S2 MB	6	-	-	$MtCO_2e$	11.41	-	-
RE100	58	24	41%	TWh	73.69	30.30	41%

5.2 Environmental validation

The environmental validation data design is well structured to diminish the effects of information issues. The methodology designed by the authors is robust. However, considering that the highest emissions, as found out by the authors, are from the utilities sector, it is a pity that “Our tests were not applied to companies in the Electricity Generation sector due to their unique nature”. Probably, some tests to diminish the effects of information issues on such data would have been relevant.

As mentioned above, this has now been improved. Although the full set of validation tests could not be applied to most of these companies even after further revision, a more thorough comparison against company reports was carried out as a substitute. We believe this improves our study's standing in relation to these companies.

Per year tests, cross-year consistency tests and renewable sourcing tests are robust and innovative methodologies. Considering the variety of emissions factors used across years by the same organisation as well as among different organisations, equation 10,11 and 12, with figure 8, are extremely interesting and very much appreciated methods to alleviate the differences.

5.3 Benchmark and progress indicators

The logical framework employed in the literature is consistent and the author's adaptation clear in the supplementary material. The description and considerations of the methodology are appropriate. Suggested improvements

Although data sets are available and usable for statistical analysis, I kindly ask for the coding scripts. I also ask for the individual files for each company, to further check on the accuracy of data collection.

The code has been made public, as well as all the company files.

All CDP HTML files will be similarly provided to reviewers.

The manuscript is really sound and valuable. However, my main concern is about the relevance of this research. The authors used a sample mainly based on services and light industry and they considered only scope 1 and scope 2 because as they rightly pointed out there are no data for scope 3. They have done a very good job with the data available but not to consider scope 3 impacts for such industries is a relevant shortcoming since their impacts are mainly in that scope. I fully know that this shortcoming cannot be solved easily, but it had to be pointed out.

We thank the reviewer for the comment. We agree that Scope 3 is a crucial part of company action. The introduction now addresses this omission better, explaining why it has a value in itself to only deal with Scope 1 and 2 emissions, and why Scope 3 is so difficult to tackle.

Nonetheless, it is possible to suggest some precise improvements to work on:

- It would be suggested to evaluate whether the following recent paper Bjørn, A., Tilsted, J.P., Addas, A. et al. Can Science-Based Targets Make the Private Sector Paris-Aligned? A Review of the Emerging Evidence. *Curr Clim Change Rep* 8, 53–69 (2022) could be useful to the introduction and discussion.

We have added Bjørn's paper to the discussion section, specifically on the need better procedures and on transparency being a prerequisite for credibility.

Introduction:

“Although emissions in this scope are often larger than Scopes 1 and 2, by definition Scope 3 emissions come from third parties in the value chain of a company, meaning targets would require influencing the behaviour of other businesses [30] that might not be members in either initiative, while Scope 1+2 are always within a company's sphere of influence”

“We conclude this paper by calling for an improved approach to the intermediate steps between target-setting and claims of progress. Initiatives must strive for a consistent and transparent disclosure processes [30], with more attention put into how renewable energy is sourced by members, and with a focus on increasing membership in developing regions and in high-emitting sectors.”

Discussion:

“Our study shows that companies are successful in meeting the goals established within these two initiatives. However, achievements may not directly translate into additional emission reductions. Several recent studies evaluating corporate action have shared concerns about the additionality of action due to low transparency [19, 30] and the lack of a robust mandatory reporting framework [18].”

“For initiatives such as the SBTi and RE100, a crucial step to increase additionality would be to move to more high-quality renewable energy sourcing methods, such as PPAs and self-generation. And anyway, more transparent disclosure and a better evaluation of accounting procedures are prerequisites for a credible contribution by companies to global emission reduction [30]. Without this step, evaluation of other metrics such as Scope 3 mitigation will remain difficult.”

- It would be suggested to further exploit figure 6 by inserting other graphs showing the differences in progress between utilities and/or energy intensive companies and the other organisations. In fact, it is extremely relevant, as the authors point out in the discussion, that 85% of all GHG emissions reduction comes from a few energy intensive organisations. To enhance the clarity of figure 6 as well as the relevancy of the manuscript, it is then suggested to make graphically visible this important result after the overall graphs already present in figure 6;

We thank the reviewer for this insight.

A new graph (6d) was added to Figure 6, disaggregating progress by initiative, and keeping emission intensive members separate. We believe this new graphic makes the importance of the results found in this study apparent, as requested.

“**d** Combined Scope 1+2 emissions of all companies subdivided by membership and keeping SBTi members in emission intensive sectors separate (i.e., Electricity Generation, Energy Intensive Industry). Scope 2 Market Based data was preferred if available. See supplementary Table 6 and Table 7 more information on performance per GHG Protocol Scope.”

- It would be suggested to make clear which data are taken from CDP, and which from other sustainability documents when CDP data were lacking. This strengthens the accuracy of environmental data collection and assures reproducibility for future studies. I asked for the individual files to see whether these specifications are present. There could be a table in supplementary material or in one file excel where there is the percentage of data taken by CDP and by the other documents.

Our methodology did not include adding the source of data at an atomic level. This is an important omission in our approach. We cannot give a specific percentages.

Fortunately, we do mention all our sources in each individual company file (new available in the data repository), as well as brief descriptions of how issues were corrected. This should aid in repeatability, even if it is not the ideal approach.

- Since the authors assess only scope 1 and scope 2, it would be appropriate to insert in the supplementary material the total amount and percentages of scope 2 GHG emissions under MB and LB approaches that are aligned with companies' targets, as well as the contribution of MB and LB data to the total GHG amount (figure 2 and figure 6). It would not change the results, but it would give higher clarity and reproducibility.

We now disclose this information in the supplementary material, as requested.

Supplementary information

“Contrary to previous studies, our analysis has a special focus on the effects of collective action in the initiatives. Regardless, the individual performance of members against their targets remains an important aspect of non-state action studies. The following table discloses member progress against the extended linearised targets seen in Figure 2, Figure 3 and Figure 6.”

Supplementary Table 7: **Disaggregation of target progress per initiative.** In the case of the SBTi progress has been broken up by scope, including Scope 2 category.

Initiative	Companies	N. on track	% on track	Target unit	Total 2019	T. on track 2019	% on track
SBTi abs. S1	63	46	73%	MtCO_{2e}	157.58	134.75	86%
SBTi abs. S2 LB	16	10	63%	MtCO_{2e}	27.62	19.19	69%
SBTi abs. S2 MB	47	36	77%	MtCO_{2e}	56.12	33.68	60%
SBTi int. S1	7	-	-	MtCO_{2e}	246.14	-	-
SBTi int. S2 LB	1	-	-	MtCO_{2e}	2.52	-	-
SBTi int. S2 MB	6	-	-	MtCO_{2e}	11.41	-	-
RE100	58	24	41%	TWh	73.69	30.30	41%

- Following the previous suggestion, it would be suggested to make clearer in the text which data between MB and LB are the most prominent in the calculation (even though indirectly it should be MB).

Another table has been added to the supplementary material detailing LB and MB data used in figure 6d, per year.

Supplementary Table 6: **Disaggregation of substantive progress by GHG Scope, in $MtCO_2e$.** The data featured in Figure 6d is separated by sector and GHG Protocol scope, including Location Based (LB) and Market Based (MB) approaches for Scope 2[4]. Due to inconsistencies in emissions reporting, MB data was preferred if available. Otherwise, LB data was used.

Sector	Scope	2015	2016	2017	2018	2019
Electricity Generation	S1	371.64	330.9	302.39	250.47	196.74
	S2 LB	11.68	9.23	3.58	3.42	2.52
	S2 MB	7.72	8.41	11.11	9.89	9.99
Energy Intensive Industry	S1	206.59	194.98	173	165.77	150.67
	S2 LB	4.82	4.76	4.69	4.48	4.01
	S2 MB	15.99	10.67	9.52	8.39	7.53
Light Industry	S1	46.91	47.32	46.02	48.09	44.83
	S2 LB	15.44	13.88	11.04	7.06	7.42
	S2 MB	45.42	46.59	40.87	41.78	36.49
Services	S1	16.39	16.48	16.3	15.38	15.14
	S2 LB	26.68	7.7	6.07	5.84	5.38
	S2 MB	29.61	44.56	38.92	36.3	31.56
Transport	S1	4.17	4.26	4.29	4.2	4.15
	S2 LB	0.09	0.07	0	0	0
	S2 MB	5.51	5.14	5.12	4.28	4.03
Totals	S1	645.72	593.93	541.99	483.92	411.53
	S2 LB	58.71	35.65	25.38	20.8	19.33
	S2 MB	104.25	115.37	105.53	100.63	89.6

- It would be suggested, if possible, to try to perform a validation test (a different one, if possible) also on electricity generation sector companies, to cover the whole sample and guarantees further robustness to data collection.

A thorough review of the utility companies was carried out. Due to the importance of these companies, a more explicit summary of data issues has been included in the database, which is referred to in the methodology section.

Clarity and context

The manuscript is clear in all its forms: manuscript's structure, content and data interpretation, methodology, language, figures and tables. The methodology and results have been provided with sufficient context and consideration of previous work. A few suggestions are already detailed in the previous section.

We thank the authors and the editor for the reporting summary.

References

The manuscript adequately builds on the previous literature. References are up to date. When references are not recent, they are nonetheless extremely relevant for the paper.

REVIEWERS' COMMENTS

Reviewer #1 (Remarks to the Author):

General comments

Round 1 comment: The clarity of the writing could be improved as it can be quite hard to follow in places. This is not a comment on the content, just on the writing style. There are in the introductory sections a few places where terms are not defined or explained and remain quite inaccessible to readers not already familiar with the initiatives being evaluated.

Round 2 comment: The additions are helpful but do not fully address the issue of clarity in writing. I would recommend a thorough copy-edit to improve the readability of the text.

On the CDP categories, it is not clear from describing the Limited and Reasonable categories, what the criteria are to classify cases as Moderate and High. As the reader, I am not clear on what "Moderate" assurance would mean. It would also be helpful (perhaps in an Appendix rather than the main text) to provide a more detailed understanding of how CDP arrives at these determinations. Does CDP use the assurance providers' statements (e.g. "negative" vs "positive" statements) as the basis for classification, or something else? This is important because the authors are relying on CDP's classification method and it is important to determine how robust this third-party methodology is in order to assess its appropriateness to this application. This also applies to the Abstract version.

On the sourcing methods definitions, in the "low additionality" sentence on U-EACs, it is not clear why short contractual periods are linked to low additionality. It may be helpful, if the authors consider it necessary, to explain why in a few words. Although the reference should suffice for the interested reader. Similarly, it is not exactly clear from the definition what Utility GPs are. It seems they are essentially guaranteed green electricity certificates sold to buyers at a green premium, although it may be helpful to explain more clearly why Utility GPs have low potential additionality, and how them being linked to public support schemes may affect their additionality.

On Table 2, the four progress indicators make sense on their own, but it is still not clear from the table how they are combined and how they combine to tell an overall story of the company's progress. The final row states "causal impact", which is not a priori a justifiable statement (this requires robust empirical evidence), nor is it clear what the measure of "overall effectiveness" is. Are the indicators weighted or combined in a particular manner that describes overall effectiveness, or are they treated discretely as qualitative components of a narrative description of company activity?

Round 1 comment: The focus on robustness of target-meeting disclosure is very helpful and may even deserve greater prominence, as it is the litmus test for whether ambition is just a marketing exercise, or has real substance. It may be worth reporting in the abstract some of the statistics on the distribution of verification quality - with the majority being classified as limited, and very little high-quality. If there is evidence available, it may be helpful to provide a little context on what makes these verifiers' quality limited and the extent to which conflicts of interest and insufficient due diligence are a problem in the disclosure industry.

Round 2 comment: The authors state that addressing conflicts of interest and lack of due diligence would require a different type of study. While this may be true, it is still relevant to this study to provide at least a cursory review of what the state of the literature and evidence on these topics is, since this has implications for the quality of the data used in this study and the reliability of the analysis conducted here.

Round 1 comment: There is ambiguity over what is meant by "additionality" in this paper. The authors seem to use this term with reference to the instruments used to procure renewable energy. What does

not seem to be addressed but should be even if the data is insufficient to provide much insight, is the extent to which companies' actions to decarbonise themselves are "additional" to what would have happened without these targets. The trends, which seem to indicate some companies' emissions falling prior to setting targets, suggest perhaps not. Since this paper is concerned with companies' success in meeting their own targets, it would be very relevant to discuss with more focus how much active change has been required for these companies to achieve them. Similarly, it would be helpful to see more critical engagement with the possibility that declining energy intensity/changing composition of business activities/declining carbon intensity of electricity have allowed companies to claim responsibility for the observed reductions where in fact they would likely have occurred anyway due to wider trends in, for example, the cost of renewable energy, the rules governing dispatch of electricity in power markets, etc. The lack of clarity over this issue means claims that the observed trends in the data "demonstrate that the target setting approaches set by each initiative are sound" (p.8) are unconvincing.

Round 2 comment: The additions to the Discussion section are useful but they are also a good example of why this paper could benefit a copy-edit, because the sentence construction can be hard to follow and there are numerous grammatical mistakes, and missing/incorrectly spelt or used words in places. It is not clear, for instance, what "a substantial part of the actions implemented" refers to - the UNEP study, or the actions in the dataset used in this study? I assume the latter but some editing would help make this clearer. It is also not quite clear why the authors suggest that the UNEP estimate of non-additionality is too conservative (although they are probably correct to do so). Is the UNEP report's threshold for additionality in the energy sector too generous in the authors' view, given the reality of low-additionality climate actions by firms that are dominated by renewable energy purchases? This may be a perfectly reasonable claim but the logic should be explained more clearly.

The authors have not really addressed the point on discounting the possibility that company activities may not be additional. This speaks to my earlier concern that the authors are conflating two different types of additionality: (i) additional renewable energy procurement and (ii) additional climate actions. Where renewable energy is the cheapest source of electricity, action to increase its procurement would satisfy (i) but not (ii), because it would have been done anyway regardless of climate concerns. I would like to see this distinction made more clearly and cleanly in the text.

Round 1 comment: Excluding Scope 3 emissions is a major omission since it comprises the bulk of many large companies' emissions. While this omission is understandable due to data limitations, the authors should be more upfront about this in the abstract and introduction, noting that their results are confined to direct emissions and emissions from energy purchases and exploring the implications of only looking at Scope 1 and 2 in more detail, rather than referring to this in the last sentence of the paper without any analysis of how this limits or qualifies the findings in the paper.

Round 2 comment: Contrary to the authors' claim in the updated introduction text, Scope 3 emissions are often within a company's sphere of influence both in terms of the inherent carbon context of their product (e.g. an oil and gas company) and their purchasing power (e.g. offsetting air travel by their employees). Perhaps it is better to say that the influence a company has over its Scope 1 and 2 emissions is more direct but it can, depending on circumstances, also have a major influence on different components of Scope 3. The point on inconsistency of Scope 3 estimates is well taken, but this also applies to Scope 1 and 2 (see SSRN paper Jia, Ranger, and Chaudhury, Designing for comparability: A foundational principle of analysis missing in carbon reporting systems. (October 25, 2022). <http://dx.doi.org/10.2139/ssrn.4258460>). The last argument (namely that if a company fails to take action on 1+2 it is unlikely to take action on 3) is perhaps the most convincing. Overall the additions address my initial comment but I would recommend some minor changes to the text based on my additional comments here.

Round 1 comment: The claim in the discussion section "that the target setting approaches set by each initiative are sound" does not necessarily follow from the results (see general comments). Why does

the fact of emissions reductions ahead of these targets demonstrate that the approach is sound? Please consider re-wording less strongly and acknowledging more clearly that there are reasons why the targets have been met that may be unrelated to the initiatives and the target setting approach itself. This is a subtle but very important nuance as this sort of phrasing can give the misleading impression that the initiatives have not only successfully corralled companies into setting targets but that this process was the only relevant factor in achieving them, which the authors have not shown convincingly. And as the discussion goes on to say on p.9, the demonstrated reductions have taken place among a handful of companies, with little evidence that the target-setting exercise has achieved anything much beyond increased renewable energy procurement, which in any case may well have happened irrespective of target-setting (see general comments on additionality). This finding, i.e. that the vast majority of emissions reductions in this analysis are due to renewable energy generation and procurement either directly, or more often through instruments of varying quality, and very little fundamental (Scope 1) or cross-value-chain (Scope 3) progress has been made on emissions, could be made more prominent as a central result of this analysis that would also make clearer the limitations of the data and climate actions taken.

Round 2 comment: See my response above regarding additionality and other potential drivers for change beyond membership of the assessed initiatives. Overall, the other changes made to the abstract, results and discussion sections address this point satisfactorily.

Specific comments:

Most of the Round 1 specific comments have been addressed, with the exception of the following.

Round 1 comment: Please [explain] how the baseline figure is calculated - is this done consistently across all companies on the basis of a single baseline year?

Round 2 comment: Please include in the updated text an explanation of what is done when the baseline exceeds historical emissions or no baseline is disclosed (i.e. 100% of historical emissions is assumed as the baseline). This makes sense in the authors' reviewer response, but needs to be added to the paper itself.

Round 1 comment: Figure 3 - the right hand panel (b) doesn't seem very necessary or additional to (a). Suggest keeping just (a).

Round 2 comment: I understand the response, but the right hand panel (b) still isn't easy to interpret. The authors may want to consider making the plot a stacked bar graph with Renewable and Non-renewable electricity (since the distance between the Renewable and Total is simply Non-renewable TWh). This may just make the figure easier to read.

Round 1 comment: Please explain more clearly why U-EACs are considered lower quality than PPAs and why their displacement of FF sources and emissions reductions attributes are different to those of PPAs. The logic is not made explicit. If the logic is indeed sound, the authors should draw attention to most of the RE procurement being of this lower quality, but also consider why some companies might choose U-EACs over PPAs: are they cheaper, for example? If so, why?

Round 2 comment: See earlier (general) comments. Explanation makes sense. It would be nice to have a sentence on why companies might buy U-EACs rather than PPAs (and whether they are systematically incentivised to use one rather than the other) but this is not essential.

Reviewer #2 (Remarks to the Author):

See attached file.

Congratulations on doing a grand job on the various points raised by reviewers. The Scope 3 discussion was especially excellent and persuasive. Likewise, the additionality observations in the conclusion were very useful and further research in this context would be insightful. I have some final points that you may wish to consider.

1. You note that reductions are concentrated in sub-sectors. Would this be expected given the marginal costs and differential policy targeting are likely to focus decarbonisation trajectories into sectors? The point you make is a valid one, and it can be explained without letting other sectors 'off the hook'. Likewise, if participants are undertaking critical scope-3 decarbonisation (lets be optimistic) then these actions would not show up in your analysis. My suggestion is to keep the point with a slight softening of the delivery.
2. An aspect that I hope you will be able to address further is your point about limited and reasonable assurance. From your description of the data issues uncovered by your work, I would suggest that limited assurance was a wise decision by the assurance providers. That we may wish to reasonable assurance may be taken for granted (if we assume such assurance is helpful) and your work highlights how far we have to travel to get there. This may be a useful point to make explicitly.

Papers such as this: <https://onlinelibrary.wiley.com/doi/10.1111/j.1835-2561.2009.00056.x> and this: <https://onlinelibrary.wiley.com/doi/10.1111/j.1099-1123.2011.00439.x> are behind the points I made on this aspect (I don't think you need to reference these but I thought it useful to show you were I am drawing my evidence base from).

3. How do you calculate an over/under representation of companies in the G500? Are you using GDP shares as the denominator or number of companies (stratified by size)? I tend to think that the list is the list ... rather than anticipate that there will an even representation across any sector, region or country. I would invite you to re think assertions about over/under representation as you don't need it to make your points. The way you articulate who is in/out on page 4 is ideal and doesn't need a hard to verify benchmark.

Reviewer #3 (Remarks to the Author):

Dear authors,

I revised your revision and I am happy with your work.

However recently has been published a new paper on Nature that has some similarities with yours. I would suggest to amend yours highlighting synergies and added value of your paper compare to that. The paper is the following:

Bjørn, A., Lloyd, S.M., Brander, M. et al. Renewable energy certificates threaten the integrity of corporate science-based targets. *Nat. Clim. Chang.* 12, 539–546 (2022).

<https://doi.org/10.1038/s41558-022-01379-5> . What is the added value of your article?

Here below my comments to your revision:

Validity

We thank you for your review. We are happy with that.

Significance

We thank you for your review. We are happy with that, but we feel that the boundary (scope 1+2) should be included also in the abstract. In this way there is no ambiguity, and the reader is immediately aware of the significance of the results.

Data gathering

We thank you for your review. We are happy with that.

Environmental validation

We thank you for your review. We are happy with that.

Benchmark and progress indicators

We thank you for your reviews. We are happy with them.

Thank you and good luck!

Round 2:

Reviewer #1 (Remarks to the Author)

General comments

Round 1 comment: The clarity of the writing could be improved as it can be quite hard to follow in places. This is not a comment on the content, just on the writing style. There are in the introductory sections a few places where terms are not defined or explained and remain quite inaccessible to readers not already familiar with the initiatives being evaluated.

Round 2 comment: The additions are helpful but do not fully address the issue of clarity in writing. I would recommend a thorough copy-edit to improve the readability of the text.

We thank the reviewer for their feedback.

We have thoroughly revised the wording in the Introduction, Results and Discussion sections to make the text easier to follow. We hope these improvements address the point raised here.

On the CDP categories, it is not clear from describing the Limited and Reasonable categories, what the criteria are to classify cases as Moderate and High. As the reader, I am not clear on what "Moderate" assurance would mean. It would also be helpful (perhaps in an Appendix rather than the main text) to provide a more detailed understanding of how CDP arrives at these determinations. Does CDP use the assurance providers' statements (e.g. "negative" vs "positive" statements) as the basis for classification, or something else? This is important because the authors are relying on CDP's classification method and it is important to determine how robust this third-party methodology is in order to assess its appropriateness to this application. This also applies to the Abstract version.

A new section has been included in the supplementary material detailing the differences between the verification categories, including a new supplementary figure subdividing Figure 4a into "None", "Negative" and "Positive" wording. We chose to keep this figure in the Supplementary Material to avoid readers misinterpreting "Negative Wording" as a failure to pass third-party verification.

Also, the main manuscript has been updated to explain the equivalency qualifications. Mentions of "Limited" verification have been updated to "low level" where applicable.

On the sourcing methods definitions, in the "low additionality" sentence on U-EACs, it is not clear why short contractual periods are linked to low additionality. It may be helpful, if the authors consider it necessary, to explain why in a few words. Although the reference should suffice for the interested reader. Similarly, it is not exactly clear from the definition what Utility GPs are. It seems they are essentially guaranteed green electricity certificates sold to buyers at a green premium, although it may be helpful to explain more clearly why Utility GPs have low potential additionality, and how them being linked to public support schemes may affect their additionality.

We thank the reviewer for the insight, and now include a brief explanation with references.

“The literature considers U-EACs and Utility GPs to have lower quality^{19,48,49} meaning they may not translate to actual displacements of fossil fuel energy sources and GHG emission reductions. U-EACs can be purchased separately from a company’s energy consumption, and are considered poor alternatives due to their low price⁵⁰, weak relation to additional renewable energy installations⁵¹ and because they do not reflect the physical flow of energy at the point of consumption²⁰. Utility GPs are contractual instruments between utilities and companies which offer lower additionality due to the plethora of government support schemes offered to utilities to increase their renewable generation and, in some cases, because they may be based on repackaged U-EACs purchased by the utility⁴⁸.”

On Table 2, the four progress indicators make sense on their own, but it is still not clear from the table how they are combined and how they combine to tell an overall story of the company's progress. The final row states "causal impact", which is not a priori a justifiable statement (this requires robust empirical evidence), nor is it clear what the measure of "overall effectiveness" is. Are the indicators weighted or combined in a particular manner that describes overall effectiveness, or are they treated discretely as qualitative components of a narrative description of company activity?

We have improved our explanation of the framework in the text:

“This framework evaluates the steps taken by companies in climate initiatives sequentially: from setting appropriate targets, to improving their capacity and implementing changes, and finally evaluating the outcome of their actions. This is done through four indicators: Ambition, Robustness, Implementation and Substantive Progress. Although each indicator is composed of quantitative metrics, all four are qualitatively combined to evaluate and contextualise the mitigation potential of the initiatives.”

In the table, we have changed “causal impact” (the default naming used in the paper by Hale et al.) to “evaluation of potential”, and improved its description to make clear that we treat the indicators discretely.

Round 1 comment: The focus on robustness of target-meeting disclosure is very helpful and may even deserve greater prominence, as it is the litmus test for whether ambition is just a marketing exercise, or has real substance. It may be worth reporting in the abstract some of the statistics on the distribution of verification quality - with the majority being classified as limited, and very little high-quality. If there is evidence available, it may be helpful to provide a little context on what makes these verifiers' quality limited and the extent to which conflicts of interest and insufficient due diligence are a problem in the disclosure industry.

Round 2 comment: The authors state that addressing conflicts of interest and lack of due diligence would require a different type of study. While this may be true, it is still relevant to this study to provide at least a cursory review of what the state of the literature and evidence on these topics is, since this has implications for the quality of the data used in this study and the reliability of the analysis conducted here.

We thank the reviewer for their comment. To address this, we added a brief mention in the limitations of the study alongside our comments on data quality.

Round 1 comment: There is ambiguity over what is meant by "additionality" in this paper. The authors seem to use this term with reference to the instruments used to procure renewable energy. What does not seem to be addressed but should be even if the data is insufficient to provide much insight, is the extent to which companies' actions to decarbonise themselves are "additional" to what would have happened without these targets. The trends, which seem to indicate some companies' emissions falling prior to setting targets, suggest perhaps not. Since this paper is concerned with companies' success in meeting their own targets, it would be very relevant to discuss with more focus how much active change has been required for these companies to achieve them. Similarly, it would be helpful to see more critical engagement with the possibility that declining energy intensity/changing composition of business activities/declining carbon intensity of electricity have allowed companies to claim responsibility for the observed reductions where in fact they would likely have occurred anyway due to wider trends in, for example, the cost of renewable energy, the rules governing dispatch of electricity in power markets, etc. The lack of clarity over this issue means claims that the observed trends in the data "demonstrate that the target setting approaches set by each initiative are sound" (p.8) are unconvincing.

Round 2 comment: The additions to the Discussion section are useful but they are also a good example of why this paper could benefit a copy-edit, because the sentence construction can be hard to follow and there are numerous grammatical mistakes, and missing/incorrectly spelt or used words in places. It is not clear, for instance, what "a substantial part of the actions implemented" refers to - the UNEP study, or the actions in the dataset used in this study? I assume the latter but some editing would help make this clearer.

We improved the Introduction, Results and Discussion sections of the paper to make them easier to follow. For example, we have re-worded the sentence mentioned above to:

“Most companies in this study sourced their renewable electricity through models associated with low additional renewable capacity, corroborating other recent SBTi evaluations²⁰. Similarly, there is a significant GHG emissions overlap between both initiatives. As a result, it can be expected that until 2019 the overlap with national pledges is higher than what was suggested by studies stating best-case predictions^{9,29}.”

It is also not quite clear why the authors suggest that the UNEP estimate of non-additionality is too conservative (although they are probably correct to do so). Is the UNEP report's threshold for additionality in the energy sector too generous in the authors' view, given the reality of low-additionality climate actions by firms that are dominated by renewable energy purchases? This may be a perfectly reasonable claim but the logic should be explained more clearly.

We have reworded the sentence to make our opinion clear. We now use the UNEP report for context, and contrast our results against newer studies.

“Recent studies have shared concerns about the additionality of corporate climate action due to low transparency^{19,36} and the lack of a robust mandatory reporting framework¹⁸. These issues were forewarned by studies expressing worries over the quality of initiatives outside the UNFCCC^{5,6}, so much so that earlier assessments of their potential for mitigation already assumed that about one third of the emission reductions achieved would not be additional to NDCs²⁸.”

Here, reference 28 is the UNEP study mentioned by the reviewer.

The authors have not really addressed the point on discounting the possibility that company activities may not be additional. This speaks to my earlier concern that the authors are conflating two different types of additionality: (i) additional renewable energy procurement and (ii) additional climate actions. Where renewable energy is the cheapest source of electricity, action to increase its procurement would satisfy (i) but not (ii), because it would have been done anyway regardless of climate concerns. I would like to see this distinction made more clearly and cleanly in the text.

We corrected our statements on additionality, always providing context on whether we speak of renewable instalments or emission reductions. See the following as an example:

“Given the weak alignment of national policies with 1.5°C pathways⁵⁸, transnational climate initiatives such as the SBTi and RE100 will likely remain an important part of global efforts. Our study shows that companies are collectively successful in meeting the goals established within these two initiatives. However, achievements may not directly translate into additional GHG emission reductions or renewable energy capacity.”

Round 1 comment: Excluding Scope 3 emissions is a major omission since it comprises the bulk of many large companies' emissions. While this omission is understandable due to data limitations, the authors should be more upfront about this in the abstract and introduction, noting that their results are confined to direct emissions and emissions from energy purchases and exploring the implications of only looking at Scope 1 and 2 in more detail, rather than referring to this in the last sentence of the paper without any analysis of how this limits or qualifies the findings in the paper.

Round 2 comment: Contrary to the authors' claim in the updated introduction text, Scope 3 emissions are often within a company's sphere of influence both in terms of the inherent carbon context of their product (e.g. an oil and gas company) and their purchasing power (e.g. offsetting air travel by their employees). Perhaps it is better to say that the influence a company has over its Scope 1 and 2 emissions is more direct but it can, depending on circumstances, also have a major influence on different components of Scope 3. The point on inconsistency of Scope 3 estimates is well taken, but this also applies to Scope 1 and 2 (see SSRN paper Jia, Ranger, and Chaudhury, Designing for comparability: A foundational principle of analysis missing in carbon reporting systems. (October 25, 2022). <http://dx.doi.org/10.2139/>). The last argument (namely that if a company fails to take action on 1+2 it is unlikely to take action on 3) is perhaps the most convincing. Overall the additions address my initial comment but I would recommend some minor changes to the text based on my additional comments here.

We thank the reviewer for bringing out this paper. We have expanded our limitations with it, and improved our introduction mentioning how Scope 3 is not always required by either initiative, and that poor Scope 1+2 action might imply equally weak Scope 3 implementation:

“We do not evaluate targets related to emissions from upstream and downstream activities (Scope 3)³¹. Although Scope 3 emissions are often larger than the other two, the obligatory primary targets of both the SBTi and RE100 solely cover Scope 1+2. Only SBTi members whose Scope 3 emissions account for more than 40% of their total emissions are required to set secondary targets for this scope³⁵, and in some cases they only require enrolling suppliers in the initiative³⁶. Generally, we consider that changes within the operational or energy purchasing behaviour of these companies

gives crucial information on the legitimacy of their commitment. If Scope 1+2 action is ineffective despite being within the direct reach of these companies it is unlikely that Scope 3 actions will fare better.”

Round 1 comment: The claim in the discussion section "that the target setting approaches set by each initiative are sound" does not necessarily follow from the results (see general comments). Why does the fact of emissions reductions ahead of these targets demonstrate that the approach is sound? Please consider re-wording less strongly and acknowledging more clearly that there are reasons why the targets have been met that may be unrelated to the initiatives and the target setting approach itself. This is a subtle but very important nuance as this sort of phrasing can give the misleading impression that the initiatives have not only successfully corralled companies into setting targets but that this process was the only relevant factor in achieving them, which the authors have not shown convincingly. And as the discussion goes on to say on p.9, the demonstrated reductions have taken place among a handful of companies, with little evidence that the target-setting exercise has achieved anything much beyond increased renewable energy procurement, which in any case may well have happened irrespective of target-setting (see general comments on additionality). This finding, i.e. that the vast majority of emissions reductions in this analysis are due to renewable energy generation and procurement either directly, or more often through instruments of varying quality, and very little fundamental (Scope 1) or cross-value-chain (Scope 3) progress has been made on emissions, could be made more prominent as a central result of this analysis that would also make clearer the limitations of the data and climate actions taken.

Round 2 comment: See my response above regarding additionality and other potential drivers for change beyond membership of the assessed initiatives. Overall, the other changes made to the abstract, results and discussion sections address this point satisfactorily.

We thank the reviewer for the comment.

Specific comments:

Most of the Round 1 specific comments have been addressed, with the exception of the following.

Round 1 comment: Please [explain] how the baseline figure is calculated - is this done consistently across all companies on the basis of a single baseline year?

Round 2 comment: Please include in the updated text an explanation of what is done when the baseline exceeds historical emissions or no baseline is disclosed (i.e. 100% of historical emissions is assumed as the baseline). This makes sense in the authors' reviewer response, but needs to be added to the paper itself.

We improved our explanation:

“To enable comparisons, we used the most recent CDP responses of members since they include a section for SBTi targets, including baseline emissions. Then, we compared these baseline emissions against historical emissions stated in prior questionnaires or reports. If a company submitted a baseline that exceeded historical trends, or did not disclose a baseline at all, it was assumed that the target covered 100% of the historical emissions of the baseline year.”

Round 1 comment: Figure 3 - the right hand panel (b) doesn't seem very necessary or additional to (a). Suggest keeping just (a).

Round 2 comment: I understand the response, but the right hand panel (b) still isn't easy to interpret. The authors may want to consider making the plot a stacked bar graph with Renewable and Non-renewable electricity (since the distance between the Renewable and Total is simply Non-renewable TWh). This may just make the figure easier to read.

We thank the reviewer for the comment. We implemented the suggestion:

Round 1 comment: Please explain more clearly why U-EACs are considered lower quality than PPAs and why their displacement of FF sources and emissions reductions attributes are different to those of PPAs. The logic is not made explicit. If the logic is indeed sound, the authors should draw attention to most of the RE procurement being of this lower quality, but also consider why some companies might choose U-EACs over PPAs: are they cheaper, for example? If so, why?

Round 2 comment: See earlier (general) comments. Explanation makes sense. It would be nice to have a sentence on why companies might buy U-EACs rather than PPAs (and whether they are systematically incentivised to use one rather than the other) but this is not essential.

The low quality of U-EACs has been expanded upon in the Implementation Indicators section:

“The literature considers U-EACs and Utility GPs to have lower quality^{19,48,49} meaning they may not translate to actual displacements of fossil fuel energy sources and GHG emission reductions. U-EACs can be purchased separately from a company’s energy consumption, and are considered poor alternatives due to their low price⁵⁰, weak relation to additional renewable energy installations⁵¹ and because they do not reflect the physical flow of energy at the point of consumption²⁰.”

Reviewer #2 (Remarks to the Author)

See attached file.

The editing fixes requested in the file have been implemented.

Reviewer #3 (Remarks to the Author)

Dear authors,

I revised your revision and I am happy with your work.

However recently has been published a new paper on Nature that has some similarities with yours. I would suggest to amend yours highlighting synergies and added value of your paper compare to that. The paper is the following:

Bjørn, A., Lloyd, S.M., Brander, M. et al. Renewable energy certificates threaten the integrity of corporate science-based targets. Nat. Clim. Chang. 12, 539–546 (2022). <https://doi.org/10.1038/>.
What is the added value of your article?

We thank the reviewer for the suggestion. We have strengthened our arguments on the lack of additionality of Unbundled Energy Attribute Certificates using the study.

Here below my comments to your revision:

Validity

We thank you for your review. We are happy with that.

Significance

We thank you for your review. We are happy with that, but we feel that the boundary (scope 1+2) should be included also in the abstract. In this way there is no ambiguity, and the reader is immediately aware of the significance of the results.

We now mention that the emission reductions correspond to Scope 1 and 2 emissions.

Data gathering

We thank you for your review. We are happy with that.

Environmental validation

We thank you for your review. We are happy with that.

Benchmark and progress indicators

We thank you for your reviews. We are happy with them.

Thank you and good luck!